



# Process-based modelling of nonharmonic internal tides using adjoint, statistical, and stochastic approaches. Part I: statistical model and analysis of observational data

Kenji Shimizu[1,2]

[1]Kobe University, 1-1 Rokkodai-cho, Nada-ku, Kobe, 657-8501, Japan
[2]RPS AAP Consulting Pty Ltd, 500 Hay Street, Subiaco, Western Australia, 6008, Australia

**Correspondence:** Kenji Shimizu (kenji.shimizu.rc@gmail.com)

**Abstract.** A substantial fraction of internal tides cannot be explained by (deterministic) harmonic analysis. The remaining nonharmonic part is considered to be caused by random oceanic variability, which modulates wave amplitudes and phases. The statistical aspects of this stochastic process have not been analysed in detail, although statistical models for similar situations are available in other fields of physics and engineering. This paper aims to develop a statistical model of the nonharmonic,

incoherent (or nonstationary) component of internal tides observed at a fixed location, and to check the model's applicability using observations. The model shows that the envelope-amplitude distribution approaches a universal form given by a generalization of the Rayleigh distribution, when waves with non-uniformly and non-identically distributed amplitudes and phases from many independent sources are superimposed. Mooring observations on the Australian North West Shelf show the applicability of the generalized Rayleigh distribution to nonharmonic vertical-mode-one to mode-four internal tides in the diurnal,

semidiurnal, and quarterdiurnal frequency bands, provided that the power spectra show the corresponding tidal peaks clearly. These results demonstrate the importance of viewing nonharmonic internal tides as the superposition of many random waves. The proposed distribution can be used for many purposes in the future, such as investigating the statistical relationship between random internal-tide amplitude and the occurrence of nonlinear internal waves, and assessing the risk of infrequent strong waves for offshore operations. The proposed statistical model also provides the basis of investigating processes and parameters

controlling nonharmonic internal-tide variance in Part II.

**Short summary**

This paper demonstrates the importance of viewing internal tides (internal waves at tidal frequencies) as the sum of many random waves, because statistical principles introduce characteristics that do not exist for the sum of a few random waves. This view leads us to the existence of a universal probability distribution for internal tides, which can be used for scientific and

engineering purposes in the future, as is the case of surface waves.



## 1 Introduction

A substantial fraction of internal tides cannot be explained by harmonic analysis (based on the superposition of sinusoids at tidal frequencies with constant amplitudes and phases). The remaining nonharmonic component is considered to be caused by the random variability of stratification and background currents, which modulate the amplitudes and phases of remotely generated internal tides. In other fields of physics and engineering, statistical models for similar situations — the superposition of waves with constant frequency modulated by a random medium — have been developed. However, the previous studies of nonharmonic, incoherent, or nonstationary internal tides have focused on the temporal aspects of the stochastic process, and the probabilistic or statistical aspects have not been considered in detail. This paper develops a statistical model of nonharmonic internal tides observed at a fixed location by adapting previous statistical models in other fields, and then checks the model's applicability to nonharmonic vertical-mode-one to mode-four internal tides in the diurnal, semidiurnal, and quarterdiurnal frequency bands on a continental shelf.

Internal tides are internal waves with tidal frequencies, primarily in the diurnal (≈24 h period) and semidiurnal (≈12 h period) bands. They have different vertical structures, or modes, and lower modes have larger propagation speeds and usually larger energies. (The internal-tide modes are referred to as "baroclinic" modes to distinguish them from the usual tides, or the 'barotropic' mode. It is customary to count the first baroclinic mode as vertical mode one, or VM1.) Internal tides are generated by the interaction of tidal currents with topographic slopes, which implies their coherence with the tide-generating forces at the generation sites. However, they gradually become incoherent (or non-phase-locked) as they propagate away from the generation sites (e.g., Rainville and Pinkel, 2006; Buijsman et al., 2017; Alford et al., 2019). This process is considered to be caused primarily by phase modulation through the variability of the wave propagation speed (Park and Watts, 2006; Rainville and Pinkel, 2006), which is in turn caused by temporally and spatially varying pycnocline heaving and advection (Zaron and Egbert, 2014; Buijsman et al., 2017). Higher modes are more susceptible to this phase modulation because their lower propagation speeds increase the relative importance of background currents (Rainville and Pinkel, 2006; Zaron and Egbert, 2014). Although the variability of internal-tide generation can be substantial (Kerry et al., 2016), the amplitude modulation is overall considered to be less important than the phase modulation (Colosi and Munk, 2006; Zaron and Egbert, 2014). However, the generation variability could be more important for higher modes and quarterdiurnal (≈6 h period) internal tides on continental shelves, because they can be excited directly by the topographic conversion and nonlinear interaction of incoherent VM1 internal tides, respectively.

Several terms are used to refer to internal tides not explained by harmonic analysis, including nonstationary internal tides (Ray and Zaron, 2011; Shriver et al., 2014; Waterhouse et al., 2018; Nelson et al., 2019; Geoffroy and Nycander, 2022), incoherent internal tides (Kerry et al., 2016; Buijsman et al., 2017), and non-phase-locked internal tides (Zaron, 2022). The term "nonstationary" internal tides appears most popular, but it is problematic in this study because we aim to develop a model for a time-independent (i.e., stationary) probability distribution of random internal tides at one location, although the randomness of internal tides increases (i.e., nonstationary) following the wave propagation. The terms "incoherent" and "non-phase-locked" internal tides are not preferred in this study for two reasons. First, the scope of this paper includes cases with




random amplitude and constant phase, although it is not the main focus. Second, these terms assume forcing or a reference
    state with fixed frequency and phase; however, it may not be applicable to quarterdiurnal and higher-mode internal tides
    considered in this paper, because they can be directly excited by incoherent VM1 internal tides without the modulation process.
    Accordingly, the term "nonharmonic" internal tides is used in this study, because it describes how the random part of internal
    tides have been defined based on in-situ observations (Waterhouse et al., 2018; Geoffroy and Nycander, 2022) and numerical
modelling (Kerry et al., 2016; Buijsman et al., 2017; Savage et al., 2020) — by subtracting harmonic internal tides from the
    total. (Note that satellite altimetry studies have relied on different methodologies because of the coarse temporal sampling. See
    Nelson et al. (2019) for details.)

    Previous studies on nonharmonic internal tides have focused on the temporal aspects assuming a wave with Gaussian-
    distributed amplitude and phase (Colosi and Munk, 2006; Zaron, 2015; Geoffroy and Nycander, 2022) but, to my knowledge,
not on the probabilistic or statistical aspects. For example, the probability density functions (PDFs) of nonharmonic internal
    tides have not been derived, although the PDF of wave amplitude provides an important basis for many purposes, as seen in the
    example of surface waves for engineering applications (e.g., Horikawa, 1978). Furthermore, it appears that the importance of
    the superposition of multiple waves has not been taken into account. Since it is well-known that internal tides at an observation
    location can consist of waves arriving from multiple sources (e.g., Rainville et al., 2010) and remote sources (e.g., Ponte and
Cornuelle, 2013), it is expected from the central limit theorem in statistics that the process becomes Gaussian as the number
    of wave sources increases. However, this Gaussian limit is different from the Gaussian process assumed in previous studies, as
    shown in this paper. This matters because the difference can affect parameters for nonharmonic internal tides estimated from
    observations. Also, the requirements for convergence to the Gaussian limit have not been investigated for nonharmonic internal
    tides.

Situations similar to nonharmonic internal tides arise in other fields of physics and engineering, such as acoustics, optics, and
    communications, in which an observed wave signal consists of multiple wave components with the same frequency but with
    random phase shifts (e.g., see the summary by Abdi et al., 2000). Surface waves are treated differently to include the random
    frequency variability (e.g., Longuet-Higgins, 1983), although early studies assumed a fixed frequency (e.g., Longuet-Higgins,
    1952). For constant amplitude and uniformly distributed phase, the problem becomes equivalent to a random walk on the two-
dimensional plane (e.g., Bennett, 1948; Abdi et al., 2000). Previous studies in these fields have developed statistical models
    applicable to a wave signal consisting of a few to many wave components with random phases (Bennett, 1948; Beckmann,
    1964; Simon, 1985; Barakat, 1988), and also with random amplitudes (Barakat, 1974; Abdi et al., 2000).

    This paper aims to develop a statistical model of nonharmonic internal tides observed at a fixed location by adapting models
    developed in other fields of physics and engineering, and then to check the model's applicability to nonharmonic internal
tides. An important aspect of the model is to consider non-uniform and non-identical probability distributions for individual
    waves, because the amplitude and phase randomness of internal tides are expected to vary with the spatial distribution of
    the sources and their distances to the observation location. Although the model is developed by adapting previous models to
    nonharmonic internal tides, the model development is not trivial because there are relatively few and scattered studies that
    considered wave components with non-uniformly and non-identically distributed phases. The statistical model is then used





to show that the envelope-amplitude distribution approaches a generalization of the Rayleigh distribution as the number of independent sources increases. The model PDFs are compared to the observed PDFs at a mooing site on the Australian North West Shelf to demonstrate the applicability of the proposed model. The model is also used to revise the common simple (or "toy") model of internal tides that has been used for observational data analysis, so that it is applicable to cases with many wave sources.

This paper is organized as follows. Section 2 describes the proposed statistical model. Computational methods and the processing of observed data are described in Section 3, and the results are shown in Section 4. Implications of the results are discussed in Section 5. This paper ends with brief conclusions in Section 6. Appendix A provides the calculation of phase-speed variance from the observations, which is used later in Part II of this study (Shimizu, Companion Paper).

## 2   Statistical model

As a theoretical model of internal tides observed at a fixed location, we consider a sinusoidal time series that has the deterministic angular frequency $\omega$, a random amplitude $A$, and a random phase lag $\Theta$. Furthermore, we assume that this signal results from the superposition of independent and non-identically distributed $N$ sinusoidal components, each of which has a random amplitude $A_j$ and a random phase lag $\Theta_j$. Then, the signal can be expressed as

$$Ae^{-\mathrm{i}\Theta}e^{\mathrm{i}\omega t} = \sum_{j=1}^{N} A_j e^{-\mathrm{i}\Theta_j}e^{\mathrm{i}\omega t}$$

$$= (X+\mathrm{i}Y)e^{\mathrm{i}\omega t} = \sum_{j=1}^{N}(X_j+\mathrm{i}Y_j)e^{\mathrm{i}\omega t}, \tag{1}$$

where $t$ is time. The Cartesian form of the complex-valued amplitude $(X,Y)$ is introduced, because both polar and Cartesian forms are necessary later. Following the convention in statistics (e.g., von Storch and Zwiers, 1999), random variables are written in upper-case letters, and the corresponding lower-case letter is used for its realization, unless otherwise stated.

   Following previous studies cited in Introduction, nonharmonic internal tides are defined by subtracting harmonic internal

tides estimated by harmonic analysis (i.e., least-squares fitting of sinusoids at the tidal frequencies). So, we consider the statistics of

$$X'+\mathrm{i}Y' = (X+\mathrm{i}Y) - (\mathrm{E}(X)+\mathrm{i}\mathrm{E}(Y))$$
$$= \sum_{j=1}^{N}\{X_j+\mathrm{i}Y_j - (\mathrm{E}(X_j)+\mathrm{i}\mathrm{E}(Y_j))\} \tag{2}$$

in this study. Hereafter, $E(\cdot)$ denotes the expected value of the argument. We write the above expression in polar form as

$$A'e^{-\mathrm{i}\Theta'} = Ae^{-\mathrm{i}\Theta} - re^{-\mathrm{i}\varphi}$$
$$= \sum_{j=1}^{N}\{A_j e^{-\mathrm{i}\Theta_j} - r_j e^{-\mathrm{i}\varphi_j}\}, \tag{3}$$



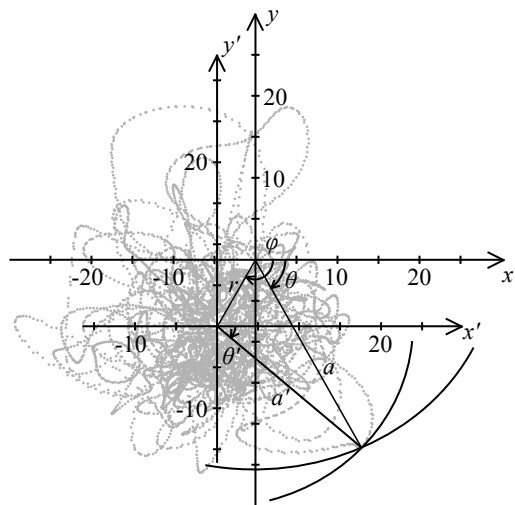

**Figure 1.** Schematics of relationships among variables used in this paper on the complex plane. $x + iy$ is total complex-valued amplitude, and $x' + iy'$ is that with zero mean. Grey dots show samples taken from nonharmonic vertical-mode-one semidiurnal internal tide at PIL200 location (described in Methods section). For illustration purposes, $r = 9$ m ($\approx$1.5 times the standard deviation of harmonic semidiurnal internal tide) and $\varphi = 120°$ are chosen arbitrarily.

where $\mathrm{E}(X) + i\mathrm{E}(Y) = re^{-i\varphi}$. Note that $r$ is the distance to the expected value of the complex vector $X + iY$ on the complex plane. Because of this, $\mathrm{E}(A'^2)$ is not $\mathrm{Var}(A)$, and $r$ and $\varphi$ are not $\mathrm{E}(A)$ and $\mathrm{E}(\Theta)$, respectively. Hereafter, $\mathrm{Var}(\cdot)$ denotes the variance of the argument. Relationships among the variables are illustrated in Fig. 1.

A particular variable of interest in this study is $A'^2$, which corresponds to the squared envelope amplitude of nonharmonic internal tide. It may not be obvious in the polar form, but provided that the individual sinusoidal components are independent, the use of Cartesian components shows that the following relationship holds generally, for non-identically distributed components, without assuming the independence of $A'_j$ and $\Theta'_j$:

$$
\begin{aligned}
\mathrm{E}(A'^2) &= \mathrm{E}(X'^2 + Y'^2) \\
&= \sum_{j=1}^{N} \mathrm{E}(X'^2_j) + \sum_{j=1}^{N} \mathrm{E}(Y'^2_j) \\
&= \sum_{j=1}^{N} \mathrm{E}(A'^2_j).
\end{aligned}
\tag{4}
$$

In this study, we refer to the second moment of $A'$ as the "total variance", and write $\sigma^2_{A'} = \mathrm{E}(A'^2)$, because it is the sum of $\mathrm{Var}(X)$ and $\mathrm{Var}(Y)$, although it is not $\mathrm{Var}(A)$.



## 2.1 General relationships

Since we consider a sum of random variables, we take the standard approach in statistics, and tackle the problem by (i) considering the joint probability density function (PDF) of $X_j$ and $Y_j$, (ii) calculating the characteristic function (i.e., the Fourier transform of the PDF) of each component, (iii) taking the product of the characteristic functions, and (iv) calculating the total PDF as the inverse Fourier transform of the total characteristic function. We first derive the relationships applicable in general in this section, and then consider specific PDFs in the following section. Because there are some pitfalls to deal with

PDFs in polar coordinates, the derivation below starts from the expression in Cartesian coordinates, although the results are written in polar coordinates. Writing the joint PDF of each component as $f_{X_jY_j}$ in Cartesian coordinates and $f_{A_j\Theta_j}$ in polar coordinates, the two are related as

$$f_{X_jY_j}(x_j,y_j)dx_jdy_j = f_{A_j\Theta_j}(a_j,\theta_j)da_jd\theta_j \tag{5}$$

in the convention in statistics (Hoyt, 1947). Note that the Jacobian of the coordinate transformation (i.e., $a_j$) is included in

$f_{A_j\Theta_j}$, so that $f_{X_jY_j} = a_j^{-1}f_{A_j\Theta_j}$ (Hoyt, 1947). This is necessary to make the integral of $f_{A_j\Theta_j}$ over the whole domain unity, and to retain the properties of PDFs (e.g., marginal and conditional probability); however, it is unfortunately a potential source of confusion. To calculate the characteristic function $\phi_j$, we consider the PDF of $(X_j',Y_j')$, define $\phi_j$ as the two-dimensional (2D) Fourier transform of $f_{X_j'Y_j'}$, and then convert the expression to its polar counterpart $(A_j,\Theta_j)$. Writing the "wavenumber" vector used in the Fourier transform as $\kappa(\cos\lambda,\sin\lambda)$, the characteristic function is:

$$\phi_j(\kappa,\lambda) = e^{-\mathrm{i}\Delta_j}\int\limits_{-\pi}^{\pi}\int\limits_{0}^{\infty}f_{A_j\Theta_j}(a_j,\theta_j)e^{\mathrm{i}\kappa a_j\cos(\lambda+\theta_j)}da_jd\theta_j, \tag{6a}$$

$$\Delta_j = \kappa r_j\cos(\lambda+\varphi_j), \tag{6b}$$

where $\Delta_j$ is the phase shift originating from the subtraction of the mean in Eq. (3). (Note that the definition of the Fourier transform follows the convention in statistics in this paper.) The total characteristic function is given by

$$\phi = \prod_{j=1}^{N}\phi_j. \tag{7}$$

For later convenience, we expand $\phi$ into the azimuthal Fourier series:

$$\phi(\kappa,\lambda) = \sum_{k=-\infty}^{\infty}\phi^{(k)}(\kappa)e^{-\mathrm{i}k\lambda}. \tag{8}$$

The total PDF is given by the inverse Fourier transform of $\phi$. We consider the inverse 2D Fourier transform in Cartesian coordinates first, and then transform the coordinates to polar coordinates, yielding

$$f_{A'\Theta'}(a,\theta) = \frac{a}{(2\pi)^2}\int\limits_{-\pi}^{\pi}\int\limits_{0}^{\infty}\phi(\kappa,\lambda)e^{-\mathrm{i}\kappa a\cos(\lambda+\theta)}\kappa d\kappa d\lambda,$$

$$= \frac{a}{2\pi}\sum_{k=-\infty}^{\infty}(-\mathrm{i})^ke^{\mathrm{i}k\theta}\int\limits_{0}^{\infty}\phi^{(k)}(\kappa)J_k(\kappa a)\kappa d\kappa, \tag{9}$$





where $J_k$ is the Bessel function of the first kind of the order $k$, and the factor $a$ is multiplied to impose Eq. (5) upon conversion to polar coordinates. The second expression is obtained using Eq. (8), the Jacobi-Anger expansion, and the properties of the Bessel function (Abramowitz and Stegun, 1972, Eqs. 9.1.5, 35, 44, and 45). Note that this total PDF is for the deviation from the mean as in Eq. (3), although the PDFs of each component $f_{A_j \Theta_j}$ in Eq. (6a) include the mean. (However, the prime is

omitted in the arguments of $f_{A'\Theta'}$ for brevity.) The radial (or envelope-amplitude) PDF is given by the marginal probability:

$$
f_{A'}(a) = \int_{-\pi}^{\pi} f_{A'\Theta'}(a,\theta)d\theta
$$

$$
= a \int_0^\infty \phi^{(0)}(\kappa) J_0(\kappa a)\kappa d\kappa. \tag{10}
$$

If $A_j$ has an upper limit $\alpha_j$, the computational load of the Hankel transform Eq. (9) and the subsequent moments can be reduced (Bennett, 1948; Barakat, 1974). This is because $A$ has the maximum value (see Fig. 1)

$$
R = \sum_{j=1}^N |\alpha_j + r_j|. \tag{11}
$$

Since the PDF is zero for $a > R$, the Hankel transform in Eq. (9) can be replaced by the Fourier-Bessel series:

$$
f_{A'\Theta'}(a,\theta) = \frac{a}{\pi R^2} \sum_{k=-\infty}^{\infty} \sum_{l=1}^{\infty} \frac{i^k}{J_{k+1}^2(j_{k,l})} \phi^{(k)}\left(\frac{j_{k,l}}{R}\right) J_k\left(\frac{j_{k,l}a}{R}\right) e^{ik\theta}, \tag{12}
$$

where $j_{k,l}$ is the $l^{\text{th}}$ root of $J_k$. Unlike Eq. (9), this requires the evaluation of $\phi^{(k)}$ only at discrete points. Then, the mean-square amplitude is given by

$$
\mathrm{E}(A'^2) = \int_0^\infty a^2 f_{A'}(a)da
$$

$$
= 4R^2 \sum_{l=1}^{\infty} \frac{1}{j_{0,l}^2 J_1(j_{0,l})^2}\left(J_0(j_{0,l}) + \left(\frac{j_{0,l}}{2} - \frac{2}{j_{0,l}}\right)J_1(j_{0,l})\right)\phi^{(0)}\left(\frac{j_{0,l}}{R}\right). \tag{13}
$$

The covariance matrix is given by

$$
\mathbf{C} = \begin{bmatrix} \sigma_X^2 & \rho_{XY}\sigma_X\sigma_Y \\ \rho_{XY}\sigma_X\sigma_Y & \sigma_Y^2 \end{bmatrix}, \tag{14}
$$

where

$$
\sigma_X^2 = \frac{1}{2}\mathrm{E}(A'^2) - \mathrm{Re}(B), \tag{15a}
$$

$$
\sigma_Y^2 = \frac{1}{2}\mathrm{E}(A'^2) + \mathrm{Re}(B), \tag{15b}
$$

$$
\rho_{XY} = \frac{1}{\sigma_X\sigma_Y}\mathrm{Im}(B), \tag{15c}
$$

$$
B = R^2 \sum_{l=1}^{\infty} \frac{1}{j_{2,l}J_3(j_{2,l})}\phi^{(2)}\left(\frac{j_{2,l}}{R}\right). \tag{15d}
$$





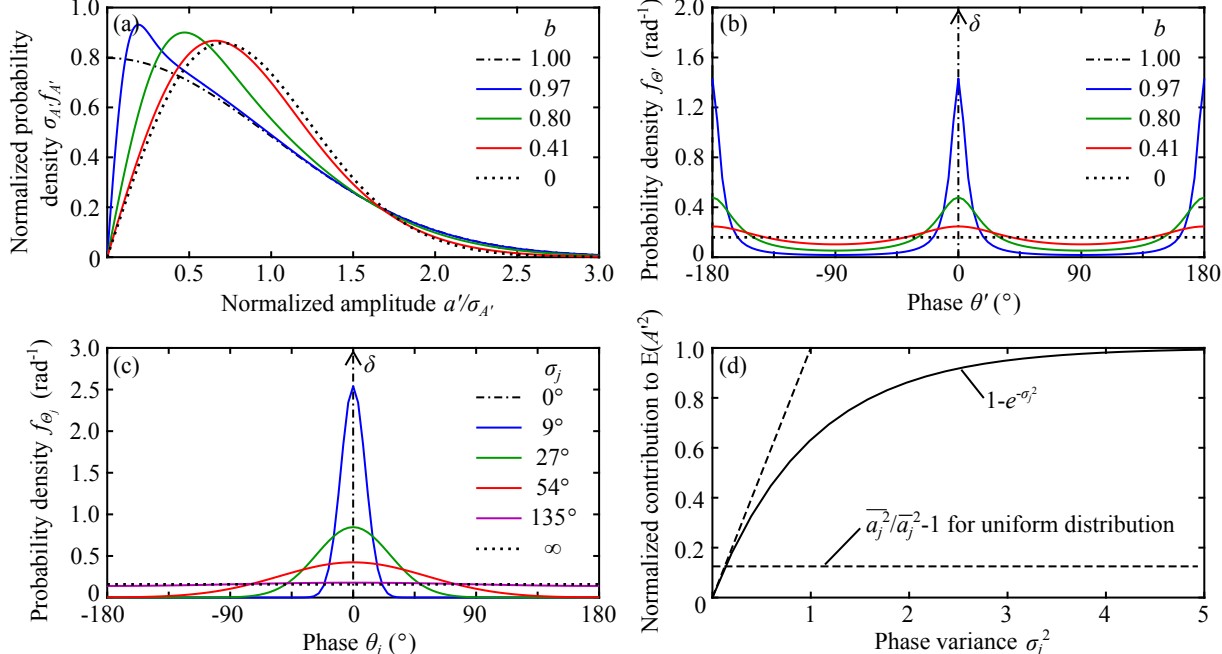

**Figure 2.** Analytic probability density functions (PDFs) used in this paper, and their properties. (a) Generalized Rayleigh distribution, Eq. (16a), (b) phase distribution of joint Gaussian distribution, Eq. (16b), plotted with $\varphi'_P = 0$, (c) wrapped normal phase distribution, Eq. (18), plotted with $\varphi_j = 0$, and (d) normalized contributions to $E(A'^2)$ (first and second terms in Eq. (20b)). In (a), amplitude and PDFs are normalized by envelope-amplitude standard deviation $\sigma_{A'}$. In (b) and (c), lines with an upward arrow indicate Dirac delta function. For sinusoidal components with equal amplitude and phase lag, distributions in (a) and (b) are limiting distributions for large $N$ under the phase distribution in (c) with the same line style.

When $N$ is sufficiently large, and $\mathrm{Var}(X_j) \ll \mathrm{Var}(X)$ and $\mathrm{Var}(Y_j) \ll \mathrm{Var}(Y)$ for all $j$ (i.e., none of the components dominate the variance), the central limit theorem states that $f_{X'Y'}$ approaches the joint Gaussian distribution (Beckmann, 1964). The corresponding amplitude and phase distributions can be calculated as marginal distributions in polar coordinates, yielding

$$f_{A'}(a) \sim \frac{2a}{\sigma_{A'}^2 \sqrt{1-b^2}} \exp\left(-\frac{a^2}{(1-b^2)\sigma_{A'}^2}\right) I_0\left(\frac{ba^2}{(1-b^2)\sigma_{A'}^2}\right), \tag{16a}$$

$$f_{\Theta'}(\theta) \sim \frac{1}{2\pi} \frac{\sqrt{1-b^2}}{1-b\cos 2(\theta - \varphi'_P)}, \tag{16b}$$

$$\sigma_{A'}^2 = \sigma_{X_P}^2 + \sigma_{Y_P}^2, \tag{16c}$$

$$b = \sigma_{A'}^{-2}\left(\sigma_{X_P}^2 - \sigma_{Y_P}^2\right). \tag{16d}$$

Here, $\sigma_{X_P}^2$ and $\sigma_{Y_P}^2 (\leq \sigma_{X_P}^2)$ are the eigenvalues of the covariance matrix Eq. (14), $\varphi'_P$ is the direction of the major orthogonal axis $x'_P$, and $I_0$ is the modified Bessel function of the first kind of the order 0. As shown by Hoyt (1947), the radial distribution function is a generalization of the Rayleigh distribution (see also Nakagami, 1960; Beckmann, 1964). The distribution becomes





the standard Rayleigh distribution when $b = 0$, and approaches a one-sided Gaussian distribution when $b \to 1$ (Fig. 2a). The

phase distribution is bimodal, and becomes uniform when $b = 0$, and two sharp peaks when $b \to 1$ (Fig. 2b). Note that $\sigma_{A'}^2 =$ $\mathrm{Var}(X) + \mathrm{Var}(Y) = E(A'^2)$ from the property of eigenvalues and Eq. (4) or Eq. (15).

## 2.2  Specific probability distribution functions

To apply the above general relationships to nonharmonic internal tides, we assume specific amplitude and phase distributions. First, we assume that the amplitude and phase variability of each sinusoidal component are independent:

$$f_{A_j \Theta_j}(a_j, \theta_j) = f_{A_j}(a_j) f_{\Theta_j}(\theta_j). \tag{17}$$

Second, we assume that $f_{\Theta_j}$ is given by the wrapped normal (or Gaussian) distribution (Mardia, 1972, p. 55)

$$f_{\Theta_j}(\theta_j) = \frac{1}{\sqrt{2\pi}\sigma_j} \sum_{k=-\infty}^{\infty} \exp\left( -\frac{(\theta_j - \varphi_j + 2\pi k)^2}{2\sigma_j^2} \right), \tag{18}$$

where $\sigma_j$ is the standard deviation of the phase, and is short-hand notation for $\sigma_{\Theta_j'}$. The wrapped normal distribution is a circular analogue of the Gaussian distribution, and defined for any one period of $2\pi$. It approaches the Gaussian distribution in

the limit $\sigma_j \to 0$, but approaches the uniform distribution in the limit $\sigma_j \to \infty$ (Fig. 2c). Note that we consider non-identical phase distribution (i.e., $\varphi_j$ and $\sigma_j$ are not necessary the same for different $j$). Then, the mean and second radial moment under Eq. (18) are given by

$$\mathrm{E}(X_j + \mathrm{i}Y_j) = r_j e^{-\mathrm{i}\varphi_j}, \tag{19a}$$

$$\mathrm{E}(A_j'^2) = \overline{a_j^2} - r_j^2 = \overline{a_j^2}\varsigma_j^2, \tag{19b}$$

where

$$r_j = \overline{a}_j e^{-\sigma_j^2/2}, \tag{20a}$$

$$\varsigma_j^2 = \left( \frac{\overline{a_j^2}}{\overline{a}_j^2} - 1 \right) + \left( 1 - e^{-\sigma_j^2} \right), \tag{20b}$$

and $\overline{a}_j = \mathrm{E}(A_j)$ and $\overline{a_j^2} = \mathrm{E}(A_j^2)$. As seen in these relationships, and as shown before by Colosi and Munk (2006), the phase spread $\sigma_j$ provides a convenient way to separate the variance of each sinusoid into the deterministic (mean) and random

(deviation) components. It is also convenient that the variance is separated into the contributions from random amplitude (the first term in $\varsigma_j^2$) and random phase (the second term). To calculate the PDF and covariance, the characteristic function of each component is needed. It is given by

$$\phi_j(\kappa, \lambda) = e^{-\mathrm{i}\Delta_j} \sum_{k=-\infty}^{\infty} \mathrm{i}^k e^{-k^2\sigma_j^2/2} e^{\mathrm{i}k(\lambda+\varphi_j)} \int_0^{\infty} f_{A_j}(a_j) J_k(\kappa a_j)\, da_j, \tag{21}$$

where $\Delta_j$ is defined in Eq. (6b). This can be substituted into Eq. (7) to calculate PDFs and moments.





We also need amplitude distribution $f_{A_j}$ to solve the problem, and we consider two contrasting amplitude PDFs. The first amplitude PDF is the constant (deterministic) distribution:

$$f_{A_j}(a_j) = \delta(a_j - \sigma_{A_j}), \tag{22}$$

where $\sigma_{A_j}$ is the constant amplitude (and $\mathrm{E}(A_j^2)^{1/2}$), and $\delta(\cdot)$ denotes the Dirac delta function. The second amplitude PDF is the uniform distribution:

$$f_{A_j}(a_j) = \begin{cases} \dfrac{a_j}{\sigma_{A_j}^2} & \text{for} \quad a_j \leq \sqrt{2}\sigma_{A_j} \\ 0 & \text{for} \quad a_j > \sqrt{2}\sigma_{A_j} \end{cases}. \tag{23}$$

This distribution is referred to as "uniform", because it corresponds to uniform probability in the radial direction between 0 and $\sqrt{2}\sigma_{A_j}$ on the $x_j$–$y_j$ plane. (Note that the factor $a_j$ comes from the requirement Eq. (5).) The distribution is normalized to have $\sigma_{A_j}^2 = \mathrm{E}(A_j^2)$, as in the constant amplitude PDF.

It is worth noting here that the relationships under the wrapped normal distribution suggest relatively small effects of amplitude distribution on the total amplitude $A'$ for two reasons. The first reason is that the contribution of random amplitude to the total variance is relatively small. The first term in Eq. (20b) is 0 (constant) and $1/8$ (uniform) for these very different amplitude distributions. In comparison, the second term can be as large as 1 (Fig. 2d) without requiring large phase spread, as pointed out by Zaron and Egbert (2014). For example, the $e$-folding standard deviation (where the dashed line reaches 1 in Fig. 2d) is 16% of the full phase $2\pi$. The second reason is that, as seen in Eq. (21), random amplitude (or smooth $f_{A_j}$) tends to smooth the characteristic function $\phi_j$ compared to the constant amplitude case. So, random amplitude tends to make the total amplitude PDF $f_{A'}$ smoother, and to make the convergence to the limiting distributions, Eq. (16), faster. For these reasons, we consider rather contrasting amplitude distributions $f_{A_j}$ in this paper.

It is also worth noting that the wrapped normal distribution is similar to the von Mises distribution used, for example, by Barakat (1988), and both distributions yield similar results within the scope of this paper. However, the two distributions are different in that the phase spread parameter in the von Mises distribution is not standard deviation and lacks clear meaning when the distribution deviates significantly from the Gaussian distribution, whereas the phase spread parameter of the wrapped normal distribution is the standard deviation, and could be estimated by various means. The wrapped normal distribution is chosen in this paper so that a stochastic model can be used to estimate the phase spread parameter in Part II of this study (Shimizu, Companion Paper).

## 3 Methods

### 3.1 Calculation of theoretical probability density function

We investigated the convergence rate of the PDFs to the limiting distributions, Eq. (16), by calculating PDFs and covariance matrices under the specific phase and amplitude distributions in Section 2.2. The azimuthal Fourier coefficients $\phi^{(k)}$ in Eq. (8) for $k = 0, 2$, and radial integration in Eq. (21) were calculated numerically.





The PDFs and covariance matrices were calculated in the Fourier-Bessel series using Eqs. (12)–(15). It is worth noting that, for large $N$, the majority of $f_{A'\Theta'}$ tend to be located in a much smaller area near the origin compared to the whole non-zero area. For example, the PDF of the generalized Rayleigh distribution becomes small for $a > 3\sigma_{A'}$. In such cases, $f_{A'\Theta'}$ excluding the tail can be evaluated by artificially reducing $R$ from Eq. (11), which can provide substantial reduction of the computational cost with a relatively small loss of accuracy. In this paper, $R = 4\sigma_{A'}$ was used for computational efficiency.

The convergence of the Fourier–Bessel series solution was slow when the PDF contained singularities, peaks, or edges. With the above choice of $R$, about 10 terms of the Fourier-Bessel series were sufficient when the resulting PDF was close to the standard Rayleigh distribution; however, more than 1000 terms could be required when the resulting PDF had sharp peaks or edges, or the $b$ parameter in Eq. (16) was small. The Fourier–Bessel series was extremely inefficient when the resulting PDF had singular points or the $b$ parameter was very close to one. Fortunately, these difficulties appear to occur only for small $N$ or almost the same $\varphi_j$. For example, for constant $A_j$ and uniformly distributed phase, singularity occurs up to $N = 4$ (Simon, 1985). In this paper, we used 1000 and 100 terms for constant and uniform amplitude cases, respectively. Cases with singularities are not considered.

## 3.2    PIL200 observations

We investigated the applicability of the proposed statistical model to nonharmonic internal tides by comparing the statistical model with measurements at the PIL200 location on the Australian North West Shelf (115.915°E, 19.435°S, water depth ≈200 m). A mooring consisting of CTDs, thermistors, and an ADCP was deployed from 20 February 2012 to 18 August 2014, as a part of the Australian Integrated Marine Observing System (IMOS). The measurements consisted of five half-yearly deployments. Although the number and heights of instruments as well as instrument settings varied over the whole measurement period, temperature and salinity were overall measured approximately at 10 and 20–30 m intervals, respectively, over the whole water column except in the upper 20–30 m. Typical sampling intervals of the CTDs and thermistors were either 60 or 120 s. Current velocity was measured at 10 m vertical intervals, and the sampling intervals varied between 300 and 1200 s among the five deployments. Pressure was measured by the ADCP located at 8–9 m above seabed.

The PIL200 data were processed as follows. We retained only data flagged as "Good_data" and "Probably_good_data", and removed suspicious salinity records. Then, we interpolated salinity to the thermistor depths, removed high-frequency variability by low-pass filtering temperature and salinity with a cut-off period of ≈1 h, sub-sampled them at 15 min intervals, and calculated isopycnal elevation. When vertical salinity interpolation was difficult because of bad or missing data at multiple levels, we did not attempt to calculate isopycnal elevation. We used isopycnal densities from 1021.00 to 1026.25 $\mathrm{kgm}^{-3}$ at 0.25 $\mathrm{kgm}^{-3}$ intervals, which resulted in roughly one isopycnal in every 10 m. Surface elevation was calculated from the pressure measurements, and then low-pass filtered and sub-sampled in the same way. Then, we calculated surface and isopycnal displacements by subtracting the corresponding background elevation, calculated by low-pass filtering the isopycnal elevation with a cut-off period of ≈62 h to remove tidal and inertial variability. Current velocity was processed similarly by removing high-frequency variability, by sub-sampling, and then by subtracting the background (low-frequency) currents.





**Figure 3.** Time series of variables related to vertical-mode-one (VM1) internal tides from PIL200 observations. (a) Celerity and low-pass filtered (subtidal) equivalent background current speed (defined in Eq. (A2b) in Appendix A), (b) maximum and surface values of VM1 structure function, (c) scaled isopycnal-displacement amplitude and its harmonic component, and (d,e) envelope amplitudes and Greenwich phase lags of diurnal, semidiurnal, and quarterdiurnal components of nonharmonic internal tides. In (a), dashed line shows least-squares fit of annual and semi-annual cycles to celerity.





### 3.3 Vertical-mode amplitude estimation

We considered vertical-mode-one (VM1) to mode-four (VM4) internal tides, whose amplitudes and energetics were estimated
as follows.

We calculated the first five modes ($\hat{\phi}_n$ for $n = 0, 1, 2, 3, 4$) and the associated celerities ($c_n$) as a function of time (at 15
min intervals) using the low-pass-filtered (background) isopycnal elevation and the formulation of vertical modes in Shimizu
(2017, 2019). Hereafter, the subscript $n$ denotes mode index, which is 0 for the barotropic mode, 1 for VM1 (the first baroclinic
mode), etc. The most common normalization of vertical modes is to set the maximum value to be 1; however, for numerically
computed vertical modes, this normalization can introduce discontinuous changes as the stratification varies over time. In this
paper, the vertical modes were normalized by setting the arbitrary norm for the barotropic mode ($\hat{h}_0$) to the water depth, and
the norms for VM1 ($\hat{h}_1$), VM2 ($\hat{h}_2$), VM3 ($\hat{h}_3$), and VM4 ($\hat{h}_4$) to 1/5, 1/17, 1/38, and 1/63 of the water depth, respectively.
The celerities of baroclinic modes showed clear seasonal variation, but the above normalization of the vertical modes kept the
extreme (minimum or maximum) value of $\hat{\phi}_n$ at about 1 (black line in Fig. 3a,b). (However, note that the depths of the extreme
varied seasonally.)

Using the vertical modes, we estimated vertical-mode amplitudes of isopycnal displacement ($\hat{\eta}_n$) and horizontal velocity
vector ($\hat{\vec{v}}_n$) based on the Gauss-Markov estimation (Wunsch, 1997). This method required estimates of the error covariance,
as well as the covariance of vertical-mode amplitudes. Following Wunsch (1997), we assumed diagonal covariance matrices.
From the high-frequency end of the power spectra of the unfiltered time series, the standard deviations of surface-displacement,
isopycnal-displacement, and horizontal-velocity errors were estimated to be $\approx$0.03 m, $\approx$3 m, and $\approx$0.04 ms$^{-1}$, respectively.
The prior estimates of vertically integrated available potential or kinetic energies contained in the first five modes were set to
1000, 1000, 500, 250, and 125 J m$^{-2}$ (the energy ratio was taken from Wunsch (1997)). Since the extreme values of $\hat{\phi}_n$ are
about one, $\hat{\eta}_n$ correspond to the maximum or minimum isopycnal displacement within the water column.

Since the measurements were made on the continental shelf, the seasonal variability of stratification affected vertical modes
and related variables substantially (e.g., black line in Fig. 3a), including $\hat{\eta}_n(t)$ (not shown). Although harmonic analysis with
multi-year-long records can determine seasonally variable harmonic internal tides, non-random seasonal variation of nonhar-
monic internal tides, which is not considered in the statistical model, would make comparisons with the proposed statistical
model more difficult. Therefore, to suppress the seasonal variability, we scaled the VM1 isopycnal-displacement amplitude as

$$\hat{\eta}_n^{\text{scaled}}(t) = \frac{c_n(t)}{c_n^{\text{ref}}} \hat{\eta}_n(t), \tag{24}$$

where $c_n^{\text{ref}}$ (=0.79 and 0.38 ms$^{-1}$ for VM1 and VM2, respectively) is the root-mean-square of $c_n(t)$.

The vertically integrated available potential energy, kinetic energy, and energy flux are given by (Shimizu, 2011)

$$P_n(t) = \frac{1}{2}\hat{\rho}\frac{c_n(t)^2}{\hat{h}_n}\hat{\eta}_n(t)^2, \tag{25a}$$

$$K_n(t) = \frac{1}{2}\hat{\rho}\hat{h}_n|\hat{\vec{v}}_n(t)|^2, \tag{25b}$$

$$\vec{J}_n(t) = \hat{\rho}c_n(t)^2\hat{\eta}_n(t)\hat{\vec{v}}_n(t), \tag{25c}$$





where $\hat{\rho}$ is the constant reference density used in vertical-mode calculation (1025 $\mathrm{kgm}^{-3}$). Since $P_n$ is given by Eq. (25a), the scaled amplitude in Eq. (24) is proportional to the square root of the available potential energy, rather than the vertical displacement of isopycnals.

Please note that the scaling Eq. (24) suppresses seasonal variability in the following analyses, but does not remove the seasonality in any way. It merely uses the fact that available potential energy showed less seasonality than isopycnal displace-

315 ments. Since the product $\hat{\phi}_n \hat{\eta}_n$ is a physically meaningful quantity that has to remain the same regardless of the scaling of vertical modes, the scaling Eq. (24) makes the scaled vertical mode (i.e., $\hat{\phi}_n^{\mathrm{scaled}} = c_n^{-1} c_n^{\mathrm{ref}} \hat{\phi}_n$) more seasonally variable. Also, note that the surface expression of internal tides showed seasonal variability with or without the scaling (Fig. 3b), which might be relevant for satellite altimetry but is not the focus of this paper.

The scaled isopycnal-displaement amplitudes $\hat{\eta}_n^{\mathrm{scaled}}(t)$ are the main variables analysed in this paper. The horizontal-velocity

amplitudes $\hat{\vec{v}}_n(t)$ are used only for obtaining some diagnostics and for some discussion.

### 3.4 Harmonic analysis

The T_TIDE package (Pawlowicz et al., 2002) was used for the traditional harmonic analysis (Foreman, 1977) to estimate harmonic internal tides. The whole record of $\hat{\eta}_n^{\mathrm{scaled}}(t)$ was used to estimate one set of harmonic constants. For consistency, we opted to use the common constituents used in the previous studies of nonharmonic internal tides (i.e., $M_2$, $S_2$, $N_2$, $K_2$, $K_1$,

$O_1$, $P_1$, $Q_1$), although the multi-year record length allowed the determination of more constituents. There were two exceptions to this. The first exception was that we included the seasonal cycle of $M_2$ and $S_2$ constituents (which are represented by the $H_1$, $H_2$, $R_2$, and $T_2$ constituents in the T_TIDE package), because a small seasonal cycle remained after the scaling Eq. (24). The second exception was that we included $M_4$, $MS_4$ and $S_4$ quarterdiurnal tides (or shallow water tides, which are overtides and compound tides of semidiurnal constituents), because the PIL200 location was on the continental shelf and the spectral

analysis, described below, showed clear quarterdiurnal peaks.

### 3.5 Nonharmonic internal tides

Nonharmonic internal tides were determined by subtracting the harmonic internal tides from $\hat{\eta}_n^{\mathrm{scaled}}(t)$. We analysed the diurnal, semidiurnal, and quarterdiurnal components. They were calculated by band-pass filtering the time series in the 21–28, 11–15, and 5.8–6.7 h bands, respectively. These bands were determined by the widths of the corresponding spectral peaks of

335 nonharmonic internal tides. The envelope amplitude $a'(t)$ of each component was estimated by first low-pass filtering the squared time series, and then multiplying the results by 2, which comes from the mean square of the sinusoidal "carrier" wave. Then, the phase lag $\theta'(t)$ was found by local least-squares fitting of a sinusoid to the time series normalized by the envelope amplitude over one period. (This method appeared to be more robust than the Hilbert transform.) The phase lag of each component was calculated as Greenwich phase lag with respect to the dominant constituent ($K_1$, $M_2$, and $M_4$ for the

340 diurnal, semidiurnal, and quarterdiurnal components, respectively). The record length of the PIL200 observations ($>2$ yr) was considered to be sufficiently long to analyse the statistics of nonharmonic internal tides on a continental shelf, although the uncertainties are relatively large as shown later.



### 3.6 Spectral analysis and estimation of cusp parameters

The power spectral density (PSD) of the total and nonharmonic internal tides were estimated by calculating the periodogram
of half-overlapping ≈85 day records ($2^{13}$ data points) of the corresponding time series with the Hamming window, averaging
them, and then converting the results to PSD. Throughout this study, PSD is defined as one-sided, defined for $0 \leq \omega < \infty$, to
be consistent with harmonic analysis.

For the goodness-of-fit test described below, the equivalent degrees of freedom (e.g., von Storch and Zwiers, 1999, chap.
17.1) of the nonharmonic internal-tide time series were required. The most straightforward way to estimate them was to use
$e$-folding decorrelation times from the shapes of so-called "cusps" in the estimated PSD, following Colosi and Munk (2006)
and Zaron (2022). These studies fit one Lorentzian spectrum above a constant background level to a frequency band containing
a cusp; however, two Lorentzian spectra were used in this paper, because a cusp covered multiple major tidal constituents, and
their frequency difference were not always negligible compared to the cusp width. This double Lorentzian spectral model is

$$g(\omega; \sigma_{A'}^2, T_\eta, \alpha, S_0) = \frac{\sigma_{A'}^2}{2\pi T_\eta} \left( \frac{\alpha}{(\omega - \omega_1)^2 + T_\eta^{-2}} + \frac{(1-\alpha)}{(\omega - \omega_2)^2 + T_\eta^{-2}} \right) + S_0, \tag{26}$$

where $\sigma_{A'}^2$ is the total variance of envelope amplitude in a cusp, $T_\eta$ is the $e$-folding decorrelation time, $\omega_1$ and $\omega_2$ are the
angular frequencies of two tidal constituents, $\alpha$ is the fraction of variance associated with the first constituent, and $S_0$ is
the background spectral level. (There are additional terms for one-sided spectra, but they are negligible for $\omega_1 T_\eta \gg 1$ and
$\omega_2 T_\eta \gg 1$.) For the diurnal, semidiurnal, and quarterdiurnal bands, the sets of ($O_1$, $K_1$), ($M_2$, $S_2$), and ($M_4$, $MS_4$) constituents
were used, respectively.

For cusps with an approximately Lorentzian form, the parameters $\sigma_{A'}^2$, $T_\eta$, $\alpha$, and $S_0$ were estimated by least-squares fitting
as follows. The most straightforward least-squares fitting turned out to be unsatisfactory because the background level $S_0$
could become unrealistically low or negative. So, we used a variant of the weighted and tapered least squares (Wunsch, 2006,
chap. 2.4.2), with a cost function similar to that used in data assimilation (Wunsch, 2006; Bennett, 2002):

$$\frac{1}{N_y}(\boldsymbol{y} - g(\boldsymbol{x};\boldsymbol{p}))^T \mathbf{R}_{nn}^{-1}(\boldsymbol{y} - g(\boldsymbol{x};\boldsymbol{p})) + \frac{1}{N_p}(\boldsymbol{p} - \boldsymbol{p}_{\text{init}})^T \mathbf{R}_{pp}^{-1}(\boldsymbol{p} - \boldsymbol{p}_{\text{init}}). \tag{27}$$

Here, the vector $\boldsymbol{y}$ contains the estimated PSD, the vector $\boldsymbol{x}$ contains $\omega$ where the PSD are estimated, and the vector $\boldsymbol{p}$ contains
the model parameters $\sigma_{A'}^2$, $T_\eta^{-1}$, $\alpha$, and $S_0$. The vector $\boldsymbol{p}_{\text{init}}$ is the initial guess of $\boldsymbol{p}$, and $\mathbf{R}_{nn}$ and $\mathbf{R}_{pp}$ are the error covariance
matrices of $(\boldsymbol{y} - g(\boldsymbol{x};\boldsymbol{p}))$ and $(\boldsymbol{p} - \boldsymbol{p}_{\text{init}})$, respectively. Diagonal $\mathbf{R}_{nn}$ and $\mathbf{R}_{pp}$ were assumed. The two terms were normalized
by the number of respective vector elements $N_y$ and $N_p$, so that varying $N_y$ for different frequency bands did not change the
relative weight of the two terms. The initial guesses of $T_\eta$ and $S_0$ were obtained by visual inspection of the estimated PSD.
Visual guesses of $T_\eta^{-1}$ were uncertain, and 1/14, 1/7, and 1/3.5 d$^{-1}$ were used as rough estimates for the diurnal, semidiurnal,
and quarterdiurnal bands, respectively. The error of $T_\eta^{-1}$ was assumed to be 50%. The errors of the estimated PSD and $S_0$
were taken from a half of the 95% confidence intervals of the spectral estimate. The initial $\sigma_{A'}^2$ was taken from the variance
of band-pass-filtered nonharmonic internal-tide time series. Since this estimate included the background level but $\sigma_{A'}^2$ does
not, the initial guess of the background level was used for its error estimate. The initial $\alpha$ was taken from the variance ratio



of harmonic internal tides in the two constituents, and the error of $\alpha$ was assumed to be 0.25. The minimum of Eq. (27) was searched numerically.

### 3.7 Estimation of model parameters

To compare the statistical model with the PIL200 observations, we applied the statistical model in Section 2 to the diurnal, semidiurnal, and quarterdiurnal frequency bands rather than to each constituent. This is because it was impractical to separate
nonharmonic internal tides into individual constituents. Although this means that the mean components, $(r, \varphi)$, vary with time due to the existence of multiple constituents, it did not cause any difficulty because harmonic tides were subtracted before analysing nonharmonic internal tides.

For the comparisons, the parameters of the PDFs in the "many source" limit, Eq. (16), were estimated from each frequency component of nonharmonic internal tides as follows. From the envelope amplitude $a'(t)$ and phase $\theta'(t)$, we first calculated
the Cartesian counterparts, $x'(t)$ and $y'(t)$, and then estimated the covariance matrix $\mathbf{C}$ in Eq. (14). The parameters $\sigma_{A'}$, $b$, and $\varphi'_p$ were calculated from the eigenvalues and eigenvectors of $\mathbf{C}$. This method appeared more robust than estimating the $b$ parameter from the skewness of the envelope-amplitude distribution $f_{A'}$.

Note that the variance of envelope amplitude $\sigma^2_{A'}$ is twice the variance of the original time series, because the sinusoidal carrier wave was removed to calculate the envelope amplitude $a'(t)$. Note also that $\sigma^2_{A'}$ estimated in this way includes the
390 background level in the PSD, which is difficult to be removed from time series data.

### 3.8 Goodness-of-fit test

The Pearson's $\chi^2$ goodness-of-fit test was used to quantitatively compare the observed PDFs with the distributions in the many source limit, because it is nonparametric and can be used with estimated parameters. To increase the reliability, the envelope amplitudes were binned with variable bin widths that correspond to equal probability under the standard Rayleigh distribution.
The phases were binned with a constant bin width. The equivalent sample size (or degrees of freedom) was calculated by dividing the record length by twice the $e$-folding decorrelation time $T_\eta$ (von Storch and Zwiers, 1999), estimated by the least-squares fitting of the double Lorentzian model to cusps in the PSD.

Note that the results of the goodness-of-fit test need to be interpreted with caution, because the statistical model for a fixed frequency is compared to the observations in the diurnal, semidiurnal, and quarterdiurnal frequency bands.





**Figure 4.** Convergence of envelope-amplitude probability density function (PDF) $f_{A'}$ with increasing number of superimposed waves $N$. Amplitude variance of individual waves $\sigma_{A_j}$ are assumed to be equal. Left and right columns show PDFs under constant and uniform amplitude distributions, respectively. First row: PDFs for phase spread $\sigma_j = 135°$ and harmonic phase lag $\varphi_j = 0$, second row: $\sigma_j = 27°$ and $\varphi_j = 0$, third row: $\sigma_j = 9°$ and $\varphi_j = 0$, and fourth row: $\sigma_j = 9°$ and $\varphi_j$ distributed evenly over $72°$. The $b$ parameter of the generalized Rayleigh distribution, Eq. (16a), is shown in each panel. Although not shown, $N = 3$ case with $\sigma_j = 9°$ and $\varphi_j$ distributed evenly over $360°$ is practically given by the generalized Rayleigh distribution. In (a), distributions have maximum amplitudes $a'/\sigma_{A'} \approx 1.9, 2.2, 2.4$ for $N = 3, 4, 5$, respectively.



## 4 Results

### 4.1 Convergence rate to the generalized Rayleigh distribution

Fig. 4 illustrates that the convergence rate of envelope-amplitude PDFs to the generalized Rayleigh distribution at the "many source" limit is faster with increasing phase spread $\sigma_j$ and more even distribution of harmonic phase lags $\varphi_j$. Considering cases with equal (constant) amplitude and harmonic phase lag ($\varphi_j = 0$), the $N = 10$ case practically reaches the limiting distribution for $\sigma_j = 135°$, but $N \approx 30$ is required for $\sigma_j = 27°$ and $9°$ (Fig. 4a-c). To see the effects of non-identical phase distribution, non-identical $\sigma_j$ and $\varphi_j$ are considered separately. If $\sigma_j^2$ is distributed linearly, the results are overall similar to the case with constant $\sigma_j$ given by the root mean of linear $\sigma_j^2$, although the results are not identical (not shown). If $\varphi_j$ are evenly distributed over $72°$ (e.g., at the intervals of $7.2°$ for $N = 10$), the $N = 10$ case practically reaches the limiting distribution for $\sigma_j = 9°$ (Fig. 4d). For the same $\sigma_j$ but with $\varphi_j$ evenly distributed over $360°$, $N = 3$ is sufficient to yield a PDF that is practically the limiting distribution (not shown). If the amplitudes are uniformly distributed with equal variance, the $N = 3$ case is reasonably close to the limiting distribution in all the $\sigma_j$ and $\varphi_j$ cases considered above (Fig. 4e-h). Also, the uniform amplitude distribution reduces the $b$ parameter, or makes the PDFs on the $x' - y'$ plane more circular (see texts in the panels). This shows that the amplitude variation tends to make the resulting PDF smoother and convergence to the limiting distribution faster, as expected in Section 2.2. Overall, the convergence rate is relatively fast.

The results here suggest that, unless observed internal tides are dominantly generated at a few generation sites, nonharmonic internal tides are likely to have PDFs close to the limiting distributions, Eq. (16), for the following three reasons: (i) it is unlikely that harmonic phase lags $\varphi_j$ are close to each other because they depend, for example, on the distance and propagation speed between the sources and the observation location, (ii) relatively small phase spread is sufficient to approach the limiting distributions, and (iii) amplitude variability tends to increase the rate of convergence to the limiting distributions. If this is the case, the universality of the total PDFs would provide a convenient basis for observational data analysis and numerical modelling; however, it would also make the analysis of the underlying processes difficult, because the total variance does not distinguish the separate contributions of individual wave components, and the total PDF (and the associated higher-order statistics) does not depend on the details of individual waves.

### 4.2 Observed time series, spectra, and energetics

The time series of harmonic and nonharmonic internal tides are shown in Fig. 3c–e. The VM1 isopycnal-displacement amplitude $\hat{\eta}_1^{\text{scaled}}(t)$ shows that the contributions of harmonic and nonharmonic internal tides are comparable at the PIL200 location (Fig. 3c). The harmonic internal tides show the spring-neap tidal cycle, but it is not clear in the nonharmonic counterpart. The envelope amplitudes of nonharmonic internal tides in the diurnal, semidiurnal, and quarterdiurnal frequency bands vary a lot without stable mean, and the phases appear random (Fig. 3d,e). These features are consistent with the PDFs at the many source limit, Eq. (16), with small $b$ parameter. The results for the higher modes are similar, except that amplitudes decrease as the mode number increases, and that harmonic internal tides are substantially smaller than the nonharmonic counterpart (not shown).

**Figure 5.** Power spectral density of (scaled) isopycnal-displacement amplitude of first four baroclinic modes. Grey shading shows 95% confidence intervals (C.I.), and green shading shows frequency bands used to define diurnal (D), semidiurnal (SD), and quarterdiurnal (QD) components, respectively. Blue solid and dashed lines show double Lorentzian spectra, Eq. (26), and their background levels, respectively, from least-squares fitting.




**Table 1.** Parameters estimated by least-squares fitting of double Lorentzian model Eq. (26) to cusps in power spectral density (Fig. 5). Background level $S_0$ in Eq. (26) is integrated over each frequency band, so that the numbers can be directly compared to potential energies in Table 2. Abbreviations are VM: vertical mode, D: diurnal, SD: semidiurnal, and QD: quarterdiurnal.

| | Decorrelation time $T_\eta$ (d) | | | Background level (J m$^{-2}$) | | |
|---|---|---|---|---|---|---|
| | D | SD | QD | D | SD | QD |
| VM1 | 10 | 10 | 4 | 5 | 6 | 4 |
| VM2 | - | 7 | 3 | - | 10 | 5 |
| VM3 | - | 6 | 3 | - | 4 | 2 |
| VM4 | - | 6 | 3 | - | 2 | 1 |

Since the PIL200 location is located on the continental shelf, there is a possibility that the local topographic excitation of higher modes by the barotropic mode or VM1 may lead to a similar behaviour of different modes. The visual inspection of the time series indeed showed intermittent periods when $\hat{\eta}_n^{\mathrm{scaled}}(t)$ for different modes were highly correlated. To check the influence of such correlation on nonharmonic internal-tide variance (the most important statistics considered in this study), the squared correlation coefficient matrix of the four modes of nonharmonic internal tides were calculated in the three frequency bands. The result shows that the off-diagonal components were mostly less than 5%, except between semidiurnal VM2 and VM3 (20%), semidiurnal VM2 and VM4 (10%), semidiurnal VM3 and VM4 (21%), and quarterdiurnal VM3 and VM4 (8%). Therefore, the influence of the correlation is considered to be small overall, and the four modes are analysed separately in the following analyses.

The power spectral density of the total and nonharmonic internal tides are shown in Fig. 5. The VM1 spectrum shows clear peaks at the diurnal, semidiurnal, and quarterdiurnal frequencies (Fig. 5a). The semidiurnal peak is tallest with $M_2$ being the dominant constituent. Since the PIL200 location is on the continental shelf, the quarterdiurnal (shallow water) internal tide is stronger than the diurnal internal tide. The subtraction of the harmonic tides reduces the heights of the peaks at the $M_2$, $S_2$, $K_1$, and $O_1$ frequencies, but otherwise makes relatively small changes to the spectrum (compare red and black lines in Fig. 5a). The spectra of the higher-mode nonharmonic internal tides show clear peaks at the semidiurnal and quarterdiurnal frequencies, but the diurnal frequency band shows either unclear or no peak (Fig. 5b–d).

The spectra show the so-called "cusp" structure around the peaks. The band-pass filters used to separate the diurnal, semidiurnal, and quarterdiurnal components were chosen based on the widths of the corresponding cusps (green shading in Fig. 5). The spectral resolution is not high enough to resolve cusps around individual tidal constituents; however, it would be difficult to separate individual cusps in any case, because the cusps are broader than the frequency differences among different constituents in the same frequency band. This provides the justification to use the diurnal, semidiurnal, and quarterdiurnal components in our analyses. The parameters associated with the cusps are shown in Table 1, and the results of least-squares fitting are shown in Fig. 5. The $e$-folding decorrelation times are 10, 6–10, and 3–4 days for the diurnal, semidiurnal, and quarterdiurnal band, respectively. These numbers are substantially smaller than those from satellite altimetry in the deep ocean (Zaron, 2022), but





**Table 2.** Vertically integrated energies (in $\mathrm{J\,m^{-2}}$) and energy fluxes (in $\mathrm{W\,m^{-1}}$) of vertical-mode-one (VM1) and mode-two (VM2) internal tides in three frequency bands. Harmonic (H) components and mean (M) and standard deviation (STD) of nonharmonic (NH) components are calculated from harmonic analysis and band-pass filtered time series, respectively.

| | Diurnal | | | Semidiurnal | | | Quarterdiurnal | | |
|---|---|---|---|---|---|---|---|---|---|
| | H | NH | | H | NH | | H | NH | |
| | | M | STD | | M | STD | | M | STD |
| VM1[†] | | | | | | | | | |
| Potential energy[+ #] | 27 | 30 | 41 | 314 | 366 | 604 | 1 | 80 | 124 |
| Kinetic energy[#] | 3 | 43 | 54 | 178 | 297 | 415 | 2 | 50 | 68 |
| Eastward energy flux | 1 | 17 | 48 | 178 | 238 | 496 | 1 | 52 | 92 |
| Northward energy flux | 3 | -5 | 41 | -132 | -203 | 546 | 0 | -57 | 105 |
| VM2[‡] | | | | | | | | | |
| Potential energy[* b] | 16 | 30 | 47 | 25 | 148 | 221 | 0 | 28 | 43 |
| Kinetic energy[b] | 3 | 30 | 38 | 27 | 85 | 121 | 1 | 21 | 30 |
| Eastward energy flux | 0 | 8 | 20 | 7 | 49 | 99 | 0 | 10 | 20 |
| Northward energy flux | 3 | -3 | 19 | -5 | -33 | 79 | 0 | -10 | 20 |

[†]To calculate standard error, divide STD for D, SD, and QD components by 8.7, 8.5, and 14, respectively.

[+]Multiply by 0.12 to convert to variance of maximum isopycnal displacement within water column, and by $8.6 \times 10^{-8}$ to that of surface displacement (neglecting seasonal cycle), in $\mathrm{m^2}$.

[‡]To calculate standard error, divide STD of SD and QD components by 11 and 16, respectively.

[*]Multiply by 0.16 to convert to variance of extreme (minimum or maximum) isopycnal displacement within water column (neglecting seasonal cycle) in $\mathrm{m^2}$.

the causes are beyond the scope of this study. The decorrelation times were used to calculate the equivalent degrees of freedom of the nonharmonic internal-tide time series.

The energetics in Table 2 shows the following results. The nonharmonic-to-harmonic variance (or potential energy) ratio is about 1.1–1.2 for the VM1 diurnal and semidiurnal components (Table 2). The VM1 quarterdiurnal component is stronger than the diurnal component and dominantly nonharmonic. This is partly expected because the nonlinear interaction of nonharmonic semidiurnal internal tides can generate nonharmonic quarterdiurnal internal tide without the modulation processes. The VM2 semidiurnal internal tide has a nonharmonic-to-harmonic variance ratio of 6, and the topographic conversion of nonharmonic VM1 semidiurnal internal tide would contribute to this large ratio. (These additional generation mechanisms are one of the major reasons why the terms "incoherent" or "non-phase-locked" tides are not used in this study.) Although the background variability seen in the PSD (Fig. 5) is included in these statistics, the comparisons of the background levels in Table 1 and the potential energies in Table 2 show that the errors are relatively small. The energy fluxes of nonharmonic VM1 and VM2




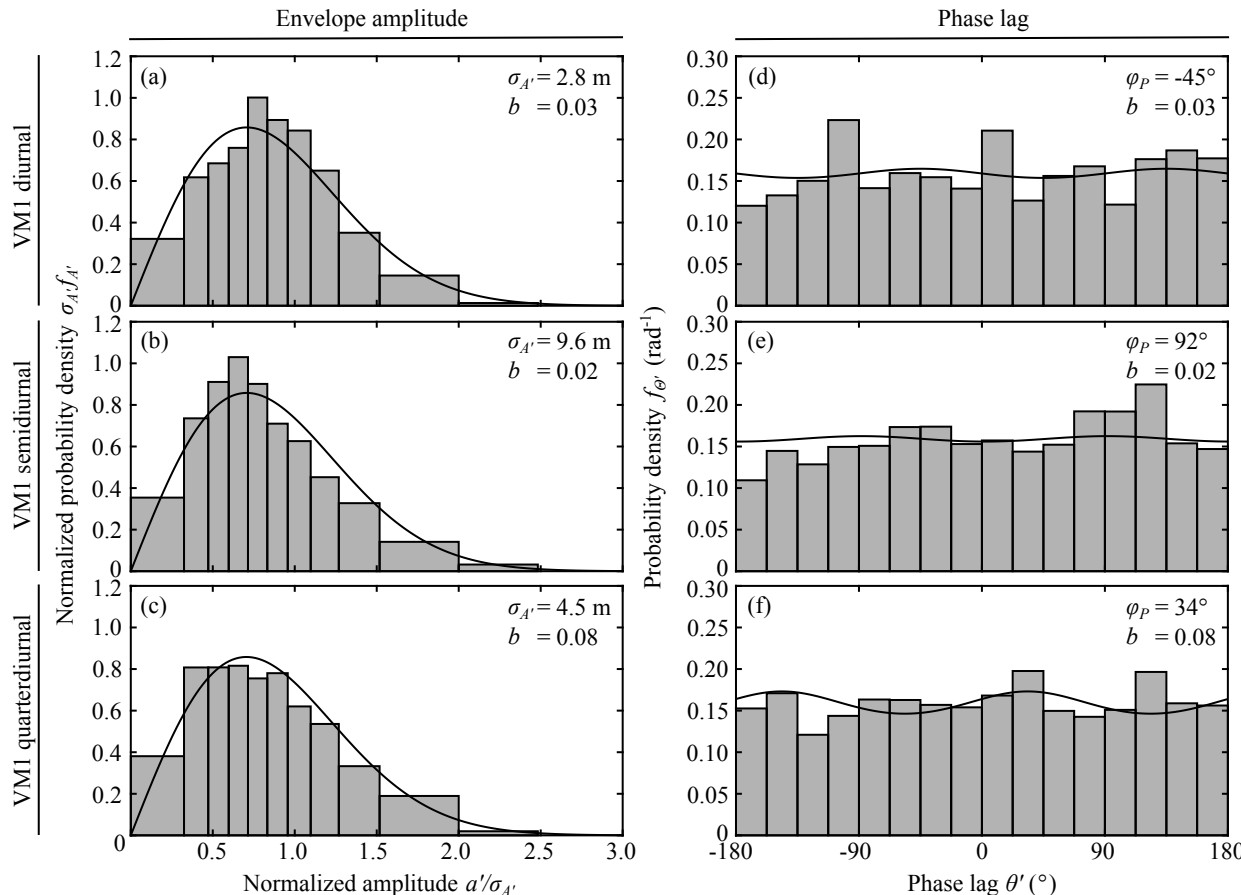

**Figure 6.** Comparisons of envelope-amplitude and phase probability density functions from the statistical model and PIL200 observations for nonharmonic vertical-mode-one internal tides. Left column: envelope amplitude, right column: phase lag. Upper, middle, and bottom rows show diurnal, semidiurnal, and quarterdiurnal components, respectively. Solid lines show distributions in "many source" limit with estimated model parameters shown in each panel.

internal tides show propagation towards ESE–SE. The ratio of the total energy and energy flux suggests that roughly half of the energy is associated with directional waves for VM1 and VM2. Note that the uncertainties of the above mean values are relatively large for nonharmonic internal tides (about ±20–30% for 95% confidence intervals after more than two years of observations), because of the highly variable nature of nonharmonic internal tides.

### 4.3 Comparisons of observed and model probability density functions

The PDFs of the envelope amplitudes and phases of nonharmonic internal tides were calculated from the corresponding time series (Fig. 3d,e for VM1).




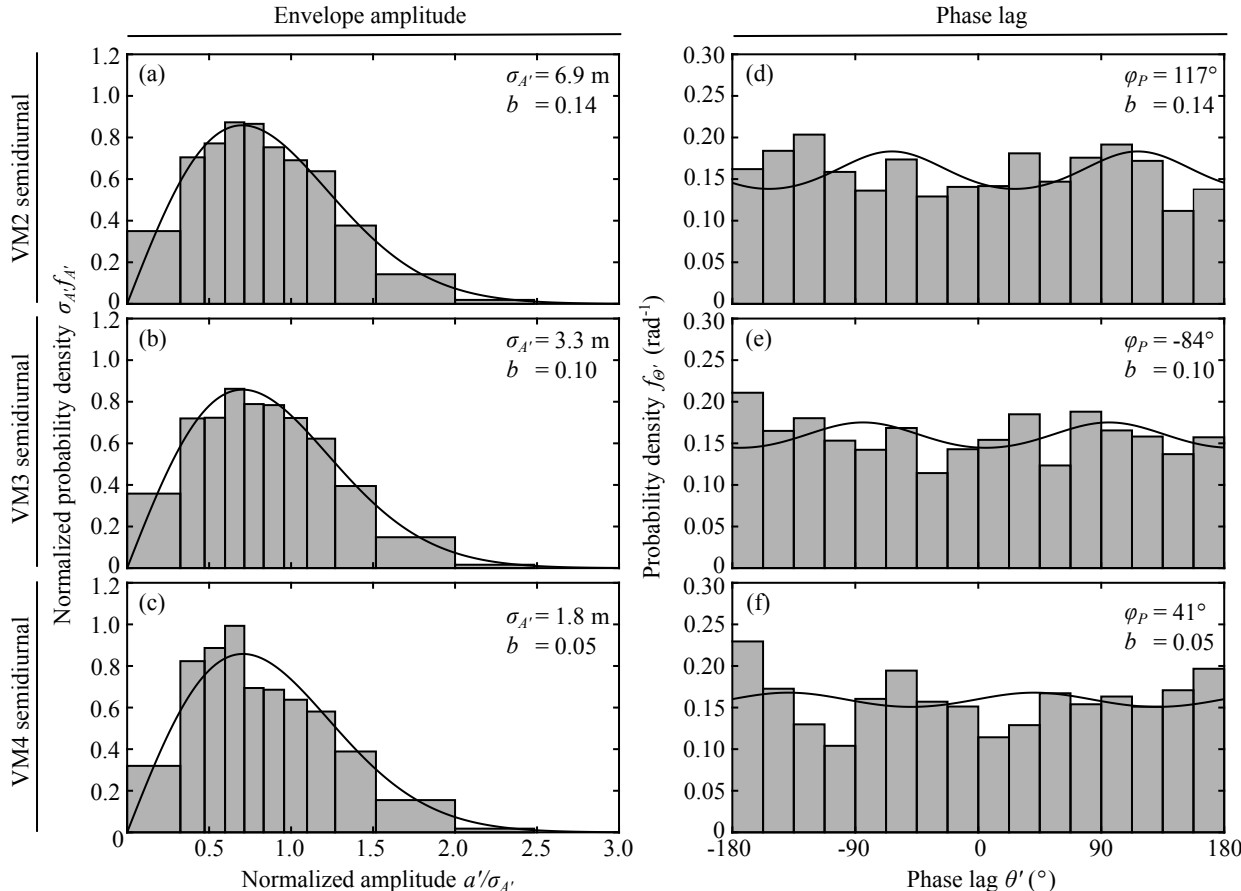

**Figure 7.** Comparisons of envelope-amplitude and phase probability density functions from the statistical model and PIL200 observations for semidiurnal frequency band. Left column: envelope amplitude, right column: phase lag. Upper, middle, and bottom rows: vertical mode two (VM2), mode three (VM3), and mode four (VM4), respectively. Solid lines show distributions in "many source" limit with estimated model parameters shown in each panel.

The comparisons of the observed and (fitted) model PDFs show that the limiting distributions, Eq. (16), provide a reasonable description of the amplitude and phase PDFs of the individual components of nonharmonic internal tides (Figs. 6 and 7). The estimated parameters are shown in the figure panels. Although the amplitude PDFs show some skewness, the phase PDFs suggest that the $b$ parameter is small. The observed and model phase PDFs may appear to disagree in some cases, because the phase PDFs were calculated as the marginal PDFs without amplitude weighting, but the model parameters were estimated based

on the covariance matrix Eq. (14), which takes amplitudes into account. However, this difference does not matter to see that the phase is roughly uniformly distributed (small $b$ parameters). For more quantitative comparisons of the PDFs, the Pearson's $\chi^2$ goodness-of-fit test shows that the observed distributions are not different from the limiting distributions at 5% significance level in all the cases in which the decorrelation time could be estimated from the cusp shapes (Figs. 6 and 7). These results





show the applicability of the proposed statistical model to nonharmonic internal tides in the many source limit, at least for the

available record length. Although the applicability is shown only at one location in this study, the convergence rate of the PDFs, shown in the previous section, suggests that the proposed statistical model has wide applicability to nonharmonic internal tides, regardless of the details of underlying physical processes. The applicability to different frequency bands and different modes, which are likely to have different generation processes, supports this speculation. (The six cases in Figs. 6 and 7 are shown to demonstrate this point, although the results look rather similar.)

## 490  5  Discussion

The major novel contributions of this paper are deriving the PDFs of nonharmonic internal tides, observationally showing their applicability, and demonstrating the importance of viewing nonharmonic internal tides as the superposition of many random waves. These contributions were made by developing a statistical model of nonharmonic or incoherent internal tides observed at a fixed location from similar models developed in other fields of physics and engineering (e.g., Barakat, 1974, 1988; Abdi

et al., 2000), and by comparing the results with the PIL200 observations. An important aspect of the statistical model is allowing non-uniform and non-identical probability distributions for individual wave components, which enables application to spatially distributed sources and increasing phase randomness with distance from the observation location.

Once the above view is adopted, some of the results of this paper might appear trivial because it follows from the central limit theorem in statistics; however, the above view was not adopted in the previous studies of nonharmonic internal tides in a

quantitative manner. A demonstration of this is the following simple model for internal tides, used by Colosi and Munk (2006), Zaron (2015), and Geoffroy and Nycander (2022):

$$\hat{\eta}_1 = (r + A')e^{i(\omega_0 t - \Theta)}. \tag{28}$$

Here, the subscript 0 is added to $\omega$ to emphasize that it is the fixed angular frequency of a harmonic tide, $r$ is the amplitude of the harmonic internal tide, and $A'$ and $\Theta$ are random amplitude and phase, respectively, which are assumed to be Gaussian.

(Although $\hat{\eta}_1$ is hereafter a random variable, it is written in lower case.) This model essentially assumes a single sinusoidal wave whose amplitude and phase are modulated by random processes, as the proposed statistical model assumes for individual wave components. However, when a nonharmonic internal tide results from the superposition of many random waves, the PDF becomes joint Gaussian in Cartesian coordinates (see Fig. 8a, or grey dots in Fig. 1 for samples from the PIL200 observations), which can be quite different from the PDF associated with the above model (i.e., $A'$ and $\Theta$ are joint Gaussian in polar coor-

dinates). The difference could be relatively minor when $\mathrm{Var}(A')$ and $\mathrm{Var}(\Theta)$ are small (Fig. 8b), but substantial when $\mathrm{Var}(\Theta)$ is not small (Fig. 8c). In particular, the above model has two awkward features when the peak of the PDF is located within a few standard deviations of the origin. First, the phase of nonharmonic internal tide can be almost uniformly distributed as seen in the right colum of Figs. 6 and 7; however, the above model becomes awkward when $\mathrm{Var}(\Theta)$ is larger than about 1, because the "wrapping" of phase is not included when the phase spread is beyond the full period $2\pi$. Second, when the PDF is seen in

Cartesian coordinates, the PDF has a peak near the origin, because the radial Gaussian distribution must be divided by radius





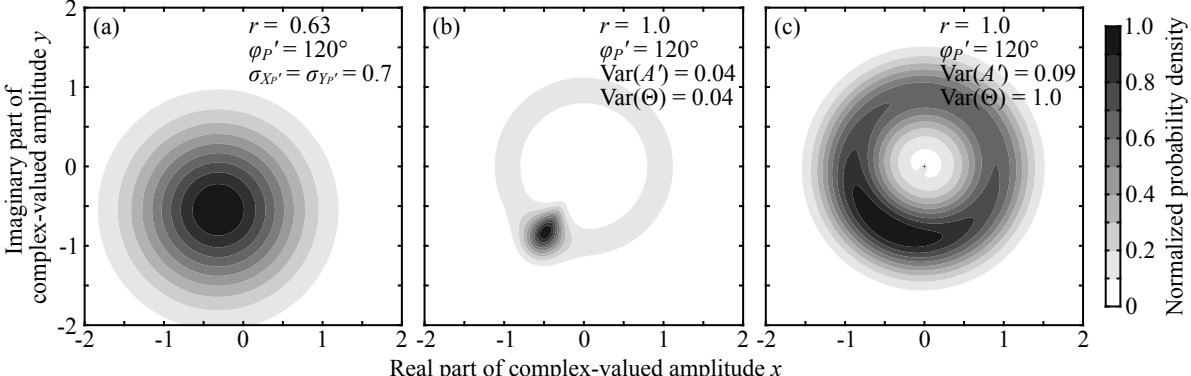

**Figure 8.** Comparison of probability density functions (PDFs) under simple (or "toy") models. (a) PDF under Eq. (29), (b) PDF under Eq. (28) with relatively small $\mathrm{Var}(A')$ and $\mathrm{Var}(\Theta)$, and (c) that under relatively large $\mathrm{Var}(A')$ and $\mathrm{Var}(\Theta)$. The parameters used are shown in each panel.

upon conversion to Cartesian coordinates to impose Eq. (5). The peak near the origin becomes wider as $\mathrm{Var}(A')$ increases. Since such a peak is unrealistic for nonharmonic internal tide, $\mathrm{Var}(A')$ effectively has a relatively small upper limit of roughly $0.1r$. Fig. 8c shows the PDF as broad as possible under these constraints. It is worth noting that $\mathrm{Var}(A')$ and $\mathrm{Var}(\Theta)$ estimated from observations in the previous studies (Colosi and Munk, 2006; Geoffroy and Nycander, 2022) are almost at these upper

limits, and that the observed distributions in Fig. 11 of Colosi and Munk (2006) appear closer to Fig. 8a than Fig. 8c.

The results of this paper suggest that the many source limit would be common in nonharmonic internal tides, and hence it would be important to construct an alternative simple model that is applicable to the joint Gaussian distribution in Cartesian coordinates. This can be done easily. Since the complex amplitude $X' + iY'$ has the joint Gaussian distribution, it appears most convenient to rotate the coordinates so that the resultant amplitudes $X'_P$ and $Y'_P$ are uncorrelated. Then, the most straightfor-

ward simple model is

$$\hat{\eta}_1 = re^{i(\omega_0 t - \varphi)} + (X'_P + iY'_P)e^{i(\omega_0 t - \varphi'_P)}, \tag{29}$$

where $\varphi'_P$ is the angle of the rotated $x'_P$ axis on the complex plane. This model is convenient because $X'_P$ and $Y'_P$ are independent Gaussian variables with zero mean, and it can deal with uniform phase distribution within the Gaussian assumption. Considering the real part of the above expression, the auto-covariance function is

$$C_\eta(\tau) = \frac{1}{2}\left\{r^2 + \left(C_{X'_P}(\tau) + C_{Y'_P}(\tau)\right)\right\}\cos\omega_0\tau, \tag{30}$$

where $\tau$ is the time lag, and $C_{X'_P}$ and $C_{Y'_P}$ are the auto-covariance functions of $X'_P$ and $Y'_P$, respectively. Following the previous studies (Colosi and Munk, 2006; Geoffroy and Nycander, 2022), we assume $C_{X'_P}(\tau) = \sigma^2_{X_P}e^{-|\tau|/T_\eta}$ and $C_{Y'_P}(\tau) = \sigma^2_{Y_P}e^{-|\tau|/T_\eta}$, where $T_\eta$ is the $e$-folding decorrelation time. Then, the Fourier transform of $C_\eta$ and appropriate scaling yield the




(one-sided) power spectral density:

$$S_\eta(\omega) = \frac{1}{2}r^2\delta(\omega-\omega_0) + \frac{\sigma_{A'}^2}{2\pi T_\eta}\left(\frac{1}{(\omega-\omega_0)^2+T_\eta^{-2}} + \frac{1}{(\omega+\omega_0)^2+T_\eta^{-2}}\right), \tag{31}$$

where Eq. (16c) is used. The last term is often omitted assuming $\omega_0 T_\eta \gg 1$, but is mathematically required for one-sided spectra (i.e., only positive $\omega$ is considered). As seen in these expressions, Eq. (29) leads to a much simpler formula of power spectral density than Eq. (28) (c.f., the derivation in Colosi and Munk, 2006).

Some readers may think that simple models such as Eqs. (28) and (29) are merely a toy model; however, the details can be important because Eq. (28) has been used for the quantitative estimation of parameters associated with nonharmonic internal tides. For example, Geoffroy and Nycander (2022) used the auto-covariance function of Eq. (28) to estimate the variance of nonharmonic internal tides from global Argo data. Another example is the estimates of the decorrelation time $T_\eta$ from satellite altimetry by Zaron (2015, 2022). Zaron (2022) fitted the Lorentzian spectrum Eq. (31) to the power spectrum of sea level anomaly, although he assumed Gaussian phase variation that does not yield the Lorentzian spectrum in general (see Colosi and Munk, 2006). If the observed nonharmonic internal tides are approximately in the many source limit, the proposed simple model and Eq. (31) would provide justification for his choice.

Note that the proposed statistical model is also applicable to a small number of wave sources, although this paper focused on the many source limit. It would be interesting to make comparisons in regions affected by a few strong sources in the future, such as around Hawaii and French Polynesian Islands (e.g., Zaron and Egbert, 2014; Buijsman et al., 2017).

Since PDFs are basic information that characterise a stochastic process, the PDFs proposed in this study can be used for many purposes in the future. For example, for surface waves, the PDF of wave amplitude is used for many engineering applications (e.g., Horikawa, 1978). Similarly, the proposed PDF can be used to assess the risk of infrequent strong waves for offshore operations. Another example would be the occurrence of nonlinear internal bores and solitary waves, which develop from internal tides. On the shallow continental shelf off California where these nonlinear waves occur regularly, Colosi et al. (2018) reported that the energy flux of internal bores and solitary waves follow the exponential distribution. If the proposed envelope-amplitude PDF is applicable to a deeper location before these nonlinear waves develop, it would allow us to investigate the statistical relationship between these nonlinear waves and the underlying internal tides.

If the many source limit is common for nonharmonic internal tides as suggested in this paper, one of the most important problems would be to understand what controls the variance of nonharmonic internal tides, because the covariance matrix Eq. (14) determines the PDF (and the associated higher-order statistics). Although the proposed statistical model includes some parameters pertaining to this point, such as the strengths of the sources and the phase spread, the comparisons with the PIL200 observations unfortunately did not provide such information. This is actually expected for any cases in which observed PDF is close to the limiting distribution, because the total variance does not distinguish the separate contributions of individual wave components, and the PDF does not depend on the details of the individual waves or the underlying physical processes. For example, the phase of observed nonharmonic internal tides can be nearly uniformly distributed when the phases of individual wave components vary less than 5% (of the total $2\pi$), and the observed amplitude tends to show large variability when the amplitudes of individual components do not vary at all. More broadly, this situation appears to be common for a



system with large degrees of freedom, as statistical mechanics shows that statistical principles make a macroscopic quantity not necessarily sensitive to the details of microscopic processes (e.g., Reif, 1965). For process-based understanding, Part II of this study (Shimizu, Companion Paper) combines the proposed statistical model with adjoint and stochastic models, which provide spatially distributed source strengths and phase spread, respectively. Then, the model suite enables us to investigate important processes and parameters controlling nonharnomic internal-tide variance.

## 6    Conclusions

This paper developed a statistical model of nonharmonic or incoherent internal tides, and compared the model probability density functions (PDFs) with the observed PDFs at PIL200 location on the Australian North West Shelf. To my knowledge, this is the first study that focused on the statistical aspects of nonharmonic internal tides, and considered the importance of viewing nonharmonic internal tides as the superposition of many random waves. The major new findings of this paper are as follows.

- The PDF of complex-valued nonharmonic internal-tide amplitude approaches the joint Gaussian distribution on the complex plane as the number of independent wave sources increases. The corresponding envelope-amplitude PDF is a generalization of the Rayleigh distribution.

- Under conditions that are likely for nonharmonic internal tides, the convergence to the "many source" limit is relatively fast. It requires about ten independent sources in most situations, and as small as three in favourable situations. This implies that nonharmonic internal tides tend to have universal PDFs.

- The observed PDFs were not different from the limiting distributions for nonharmonic vertical-mode-one to mode-four internal tides in the diurnal, semidiurnal, and quarterdiurnal frequency bands at 5% significance level, provided that the power spectra show the corresponding tidal peaks clearly. This observationally shows the applicability of the proposed PDFs in the many source limit.

- The convergence to the universal PDFs unfortunately makes process investigation based on observations more difficult, because the total variance does not distinguish the separate contributions of individual wave sources, and the PDFs become insensitive to the details of individual waves or the underlying physical processes.

Also, the statistical model was used to revise the common simple (or "toy") model of internal tides that has been used for observational data analysis, so that it is applicable to the many source limit. Since the last point above makes process investigation difficult, Part II of this study (Shimizu, Companion Paper) develops a new modelling framework and model suite to investigate important processes and parameters controlling nonharnomic internal-tide variance, based on the proposed statistical model.



*Data availability.* The PIL200 data are publicly available from https://portal.aodn.org.au/. The processed version of the observational data are available from Shimizu (2024). (This data set can be accessed only by referees until the acceptance of this manuscript. Access instruction was provided to the editor.) The statistical modelling was conducted semi-analytically using the equations presented in this paper.

## Appendix A: Calculation of phase-speed variance

As already pointed out in previous studies (e.g., Zaron and Egbert, 2014; Buijsman et al., 2017) and as shown in Part II of this study (Shimizu, Companion Paper), the variance of phase speed is an important parameter to understand and model nonharmonic internal tides. This appendix describes the calculation of phase-speed variance $\sigma_C^2$ from the PIL200 data, which is used later in Part II.

  To estimate $\sigma_C^2$, we consider the dispersion relationship of internal tides with a single vertical-mode structure under small,
random, non-tidal background isopycnal displacements and currents. Assuming waves of $\exp\{-\mathrm{i}(kx + ly - \omega t)\}$ form and neglecting the horizontal gradients of background conditions, the governing equations for linear internal tides are obtained by adding the Coriolis terms but neglecting the nonhydrostatic terms in Eq. (21) in Shimizu (2017). This yields

$$\mathrm{i}\omega\hat{\eta}_n = \mathrm{i}(\hat{h}_n + \hat{H}_n^{nt})(k\hat{u}_n + l\hat{v}_n) + \mathrm{i}(kU_n^{nt} + lV_n^{nt})\hat{\eta}_n, \tag{A1a}$$

$$\mathrm{i}\omega\hat{u}_n = \mathrm{i}\frac{c_n^2}{\hat{h}_n}k\hat{\eta}_n + \mathrm{i}(kU_n^{nt} + lV_n^{nt})\hat{u}_n + f\hat{v}_n, \tag{A1b}$$

$$\mathrm{i}\omega\hat{v}_n = \mathrm{i}\frac{c_n^2}{\hat{h}_n}l\hat{\eta}_n + \mathrm{i}(kU_n^{nt} + lV_n^{nt})\hat{v}_n - f\hat{u}_n, \tag{A1c}$$

where $\hat{\eta}_n$ and $\overrightarrow{\hat{v}}_n = (\hat{u}_n, \hat{v}_n)$ are the $n^{\text{th}}$-mode (wave) amplitudes of isopycnal displacement and horizontal velocity, respectively. Unlike the main body of this paper, $(k,l)$ and $f$ denote the wavenumber and the Coriolis parameter in this appendix, respectively. The variables $\hat{H}_n$ and $\overrightarrow{V}_n = (U_n, V_n)$ are equivalent background conditions for the $n^{\text{th}}$ mode in the nonlinear terms, defined as

$$\hat{H}_n = \sum_m \hat{N}_{nmn}\hat{\eta}_m, \tag{A2a}$$

$$\overrightarrow{V}_n = \sum_m \hat{N}_{mnn}\overrightarrow{\hat{v}}_m, \tag{A2b}$$

where $\hat{N}_{nmn}$ and $\hat{N}_{mnn}$ are the nonlinear interaction coefficients defined in Shimizu (2017), Shimizu (2019), and Appendix A in Part II, and the superscript $nt$ is used to denote the random, non-tidal version of the variable. Then, from Eq. (A1), we get the dispersion relationship:

$$(\omega - kU_n^{nt} - lV_n^{nt})^2 = f^2 + \frac{c_n^2}{\hat{h}_n}(\hat{h}_n + \hat{H}_n^{nt})\kappa^2, \tag{A3}$$

where $\kappa^2 = k^2 + l^2$. Now, we assume that the phase speed $c = \omega/\kappa$ and the celerity $c_n$ also contain random components, and derive the variance equation from the above dispersion relationship. Assuming relatively small random components, this yields

$$\frac{\sigma_C^2}{\overline{c}^2} \sim \frac{\overline{c_n}^2}{\overline{c}^4}\sigma_{C_n}^2 + \frac{\sigma_{|\overrightarrow{V}_n^{nt}|}^2}{\overline{c}^2} + \frac{1}{4}\frac{\overline{c_n}^4}{\overline{c}^4}\frac{\hat{\sigma}_{H_n^{nt}}^2}{\hat{h}_n^2}, \tag{A4}$$



where $\overline{c}$ is the mean phase speed, and $\sigma_{|\overrightarrow{V}_n^{nt}|}$ and $\hat{\sigma}_{H_n^{nt}}$ are the standard deviation of $\sqrt{(U_n^{nt})^2 + (V_n^{nt})^2}$ and $\hat{H}_n^{nt}$, respectively.

Theoretically, the second term on the right-hand side should be based on background velocity in the direction of wave propagation; however, current speed without directionality is used for simplicity. (As mentioned in the main body of this paper, the mean energetics in Table 2 suggests roughly half of the total energy in nonharmonic VM1 and VM2 internal tides is associated with directional waves, with large uncertainty.) Note that Eqs. (A1)–(A4) are applicable to the barotropic mode ($n = 0$), and hence analogous to the corresponding relationships for long surface waves governed by the shallow water equations.

The above derivation follows the idea by Zaron and Egbert (2014), but Eq. (A4) is different from their Eq. (6), which has been used, for example, by Savage et al. (2020), in three aspects. First, there is a mistake regarding $c_p$ and $c_0$ in the denominator in their Eq. (6), which is revised in Eq. (A4). Second, Zaron and Egbert (2014) added small deviations due to $c_1$, subtidal current, and background vorticity to calculate the deviation of phase speed, but the squared deviations are added in the above relationship because phase-speed variance is required in Part II. Third, the background vorticity term is omitted but the

background isopycnal-displacement term is included in Eq. (A4). Note that the variability of wave propagation paths (Park and Watts, 2006; Rainville and Pinkel, 2006) is neglected in the above argument and in Zaron and Egbert (2014). Buijsman et al. (2017) concluded that the resultant error is relatively small in the equatorial Pacific.

The phase-speed variance $\sigma_C^2$ for VM1 was estimated as follows. The variance of $c_1$ was calculated after subtracting the annual and semi-annual cycles (solid minus dashed black line in Fig. 3a), because the seasonal cycle is largely deterministic

and presumably leads to the excitation of annual and semi-annual harmonics of the major harmonic constituents. This yielded $\sigma_{C_1}^2 \approx 2.7 \times 10^{-3}$ m$^2$ s$^{-2}$. The equivalent non-tidal background displacement $\hat{H}_1^{nt}$ was calculated from Eq. (A2) as follows. First, the variable $\hat{H}_1$ was calculated using the observed nonharmonic time series of the displacement amplitudes (without band-pass filtering) as $\hat{\eta}_m$, and using $\hat{N}_{1m1}$ without the annual and semi-annual cycles. Since $\hat{H}_1^{nt}$ is assumed to be non-tidal but $\hat{H}_1$ contained nonharmonic internal tides, the variance associated with the cusps (if present) was estimated from

the spectrum of $\hat{H}_1$ by the least-squares fitting of the double Lorentzian model Eq. (26) as explained in Section 3.6, and the resultant variance was subtracted from the variance of $\hat{H}_1$ to obtain $\hat{\sigma}_{H_1^{nt}}^2$. This yielded $\hat{\sigma}_{H_1^{nt}}^2/\hat{h}_1^2 \approx 6.7 \times 10^{-3}$. The equivalent non-tidal background current speed $|\overrightarrow{V}_1^{nt}|$ was calculated in the same way, except that the low-frequency currents (less than $\approx 62$ h period) were also included. This is because background currents were neglected in the calculation of $c_1$. This yielded $\sigma_{|\overrightarrow{V}_1^{nt}|}^2 \approx 8.4 \times 10^{-3}$ m$^2$s$^{-2}$. (Since this spectrum-based method cannot be used to calculate the time series of equivalent non-

tidal background currents, low-pass filtering was used to indicate the variability of $|\overrightarrow{V}_1^{nt}|$ in Fig. 3a.) The vorticity term in Zaron and Egbert (2014) was neglected because it was not possible to estimate vorticity from the single-mooring observations. Then, for VM1, $\sigma_C^2 \approx 1.2 \times 10^{-2}$ m$^2$s$^{-2}$, or $\sigma_C$ was 10–12% of the phase speed for the three frequency bands. Note that Kunze (1985) and Zaron and Egbert (2014) did not include the contribution of background isopycnal displacements to phase speed, but it has 6–9% contribution to the phase speed variance in this example. Presumably, the relatively large contributions

of background currents and isopycnal displacements result from the relatively shallow water depth at the PIL200 location.

In Part II, $\sigma_C^2$ for higher modes are also needed. Applying the same procedure yielded $\sigma_C^2 \approx 9.5, 8.2,$ and $8.2 \times 10^{-3}$ m$^2$s$^{-2}$ for semidiurnal VM2, VM3, and VM4, respectively. The background current is the dominant (>90%) contributor in these cases.





Note that $\sigma^2_{|\vec{V}^{nt}_n|}$ and $\hat{\sigma}^2_{H^{nt}_n}$ calculated above include contributions from inertial and super-tidal frequencies. It was imprac-
tical to exclude the inertial contribution because the spectra did not show narrow inertial peaks, although the spectral level
was elevated near the inertial period (qualitatively similar to Fig. 5). The inclusion of super-tidal frequencies might appear
questionable, because the widths of the cusps appear to suggest modulation by low-frequency processes. However, this choice
was made for the following two reasons. First, as seen in the well-known example of random walk or Brownian motion, the
accumulation of high-frequency random fluctuation can produce low-frequency fluctuation. This is relevant for nonharmonic
internal tides because their random phase spread is expected to result from the accumulation of phase-speed fluctuation along
the wave propagation path (formulated as a stochastic model in Part II). Second, statistical and stochastic models usually use
the variance of random variables without frequency cut-off, even when the randomness has a clear time or length scale. For
example, $\sigma^2_{A'}$ in Eq. (31) is the variance over all frequencies, although the process has the decorrelation time $T_\eta$. So, applying
frequency cut-off could result in substantial underestimation of random phase-speed variability, and the ensuing phase spread
in statistical or stochastic analysis and modelling. For example, the contributions of frequency components lower and higher
than $\approx 62$ h period to the total $\sigma^2_{|\vec{V}^{nt}_1|}$ are about 60 and 40%, respectively. Neglecting this high-frequency component of $\sigma^2_{|\vec{V}^{nt}_1|}$
and $\hat{\sigma}^2_{H^{nt}_1}$ would result in more than 40% underestimation of $\sigma^2_C$ for VM1.

*Author contributions.* KS conceptualized this research, developed the methodology, conducted the formal analysis, investigation, and vali-
dation, visualized the results, and wrote and revised this paper.

*Competing interests.* The author has the following competing interests: Dr. Matt Rayson (editor) and other oceanographers at University of
Western Australia are involved in an on-going collaborative project with my company (involving myself) on the topic of this manuscript.
Also, the author has a competitive relationship with them for industry-funded projects on topics related to this manuscript (in Australia).

*Acknowledgements.* The PIL200 data were sourced from the Australian Integrated Marine Observing System (IMOS) – IMOS is a national
collaborative research infrastructure, supported by the Australian Government. KS thanks Steve Buchan for proof reading.



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
