# Peer review of "Process-based modelling of nonharmonic internal tides using adjoint, statistical, and stochastic approaches. Part I: statistical model and analysis of observational data"

_EGUsphere, 2024_

## Author Response (AR1)

[Note: Referee's comments are in *italic*, my replies are in Roman, and texts taken from the manuscript are "quoted". Line numbers in my replies correspond to those in the marked revised manuscript.]

-------------------- Additional author's notes to the handling editor --------------------
Dear Dr. Julian Mak,

Thank you for handling the manuscript. I have revised the manuscript carefully following the referees' comments, and double-checked the equations. Please see my point-by-point replies to individual referees' comments listed below. Other than revisions based on the referees' comments, I have made the following minor changes.

- In Eq. (19), the variance fraction $\alpha$ is changed $\beta$, because $\alpha$ is now used as the amplitudes of the Fourier--Bessel series.

- During the review stage, I have received unusual requests (rather demands) from University of Western Australia through my employer regarding Competing Interests and Acknowledgements sections. I have provided some details that are known to me, in Competing Interests section in the revised manuscript. In order to decline the requests that are morally and ethically questionable, I needed to remove the name of my employer from the manuscript completely. (However, I still clearly state that I work as a consultant in Australia in Competing Interests section.) I hope this is acceptable, but please let me know if Ocean Science prefers to deal with this matter differently.

I am looking forward to hearing the outcome of this peer-review process.

Kind regards,
Kenji Shimizu

<Author's changes in manuscript>
- Author affiliation: RPS AAP Consulting Pty Ltd was removed.
- Eq. (19), l.349, 353, 359, 367, 368: ¥alpha was changed to ¥beta.
- Competing Interests: The reason why I needed to remove the name of my employer was added.
- Acknowledgements: Acknowledgements to the referees and the handling editor were added.

-------------------- Author's reply to RC1 --------------------

*<Referee comments>*
*A new stochastic model for internal tides is suggested. It is obtained by adding different independent stochastic waves with the same frequency. The waves are represented by stochastic complex amplitudes that can easily be added. In the limit of many waves the the distribution of both the real and the complex part are Gaussian. A transformation to polar coordinates then gives a generalized Rayleigh distribution for the total amplitude (the absolute magnitude of the complex amplitude). This distribution is found to agree with a two-year time series from a mooring. This is a well-written manuscript with new and important results, which should be published. I have two major comments that the author should consider.*

Thank you for your time and effort in reviewing this manuscript, and for your constructive comments and suggestions. I have revised the manuscript following your suggestions. In particular, I have moved Section 2.1 to appendix, and wrote new Section 2.1 based on your derivation. My point-by-point replies are listed below.

*1. The theory in section 2.1 is unnecessarily complicated, which will likely deter many researchers rom reading the paper. I am not familiar with the 'standard approach in statistics' used (I am not a statistician), and I was unable to understand this section. However, the central result, eq. (16), can be obtained in a much more elementary way, as follows. By the central limit theorem, the distribution functions for the two complex components are*

$$f_{X'} = \frac{1}{\sqrt{2\pi}\sigma_{X'}} \exp\left(-\frac{X'^2}{2\sigma_{X'}^2}\right)$$

$$f_{Y'} = \frac{1}{\sqrt{2\pi}\sigma_{Y'}} \exp\left(-\frac{Y'^2}{2\sigma_{Y'}^2}\right)$$

*The joint PDF has an elliptic shape on the xy-plane. If X' and Y ' are uncorrelated, the axes of the ellipse are parallel to the x-axis and the y-axis. If not, we can find two linear combinations $X_P$ and $Y_P$ , with $X_P$ along the major axis and $Y_P$ along the minor axis if the ellipses. The joint distribution function is then*

$$f_{X'_P Y'_P} = \frac{1}{2\pi\sigma_{X_P}\sigma_{Y_P}} \exp\left(-\frac{X_P^2}{2\sigma_{X_P}^2} - \frac{Y_P^2}{2\sigma_{Y_P}^2}\right)$$

$$= \frac{1}{2\pi\sigma_{X_P}\sigma_{Y_P}} \exp\left(-\frac{a^2}{4}\left(\frac{1+2\cos\theta}{\sigma_{X_P}^2} - \frac{1-2\cos\theta}{\sigma_{Y_P}^2}\right)\right)$$

*where we used polar coordinates:*

$$x' = a \cos \theta$$

$$y' = a \sin \theta$$

*The distribution function for a is obtained by integration over θ:*

$$f_A(a) = \int_0^{2\pi} f_{X'_P Y'_P} \, d\theta$$

$$= \frac{a}{2\pi \sigma_{X_P} \sigma_{Y_P}} e^{-\frac{a^2}{4}\left(\frac{1}{\sigma_{X_P}^2}+\frac{1}{\sigma_{Y_P}^2}\right)} \int_0^{2\pi} e^{-\frac{a^2 \cos(2\theta)}{4}\left(\frac{1}{\sigma_{X_P}^2}-\frac{1}{\sigma_{Y_P}^2}\right)} \, d\theta$$

*Using the relation*

$$\int_0^{2\pi} e^{-x\cos\theta} \, d\theta = 2\pi I_0(x)$$

*we obtain*

$$f_A(a) = \frac{a}{\sigma_{X_P} \sigma_{Y_P}} e^{-\frac{a^2}{4}\left(\frac{1}{\sigma_{X_P}^2}+\frac{1}{\sigma_{Y_P}^2}\right)} I_0\left(-\frac{a^2}{4}\left(\frac{1}{\sigma_{Y_P}^2}-\frac{1}{\sigma_{X_P}^2}\right)\right)$$

*which is eq. (16a) in the manuscript. Equation (16b) is obtained by integrating over a instead of over θ.*

*I strongly suggest the author to use a derivation along these lines. If the more advanced derivation in the present manuscript provides additional important information, it may be given in an appendix, together with a motivation of why this is the case.*

<Author's response>

Thank you for your constructive suggestion. I am not a statistician either, but had to study statistics to make this paper (hopefully) acceptable for statistician. I have moved most of the material in Section 2.1 and Section 3.1 to new Appendix A, and wrote new Section 2.1 based on your derivation. The reasons I kept the original Section 2.1 in Appendix A are:

- It is required to investigate the convergence rate of the PDFs to the limiting distributions (Section 4.1); and
- There are regions in the world ocean where internal tides appear to be dominantly generated by a small number of sources. The original derivation is required to deal with cases with small to moderate number of wave sources. This is mentioned in Discussion.

Following the spirit of your suggestion, I have also added formulae to calculate covariance matrix under the wrapped normal phase distribution (Eqs. 13c-e). This allows the calculation of the *b* parameter in the limiting distribution without using the more advanced theory in Appendix A.

Also, following the comments from another anonymous referee, I have added more details, particularly the derivation of the Fourier–Bessel solution, in Appendix A, because the section is now in appendix. I hope this is acceptable.

<Author's changes in manuscript>
- l.136: New Section 2.1. "Probability distribution functions in many source limit" was added.
- l.197: Eqs. (13c)-(13e) were added.
- l.596: Most of the original Sections 2.1 and 3.1 were moved to new Appendix A.

*<Referee comments>*
*2. The material in Appendix A is unnecessary. It is barely mentioned in the main text, and the results are not used there. It seems to be connected more to Part 2 (which I have not read), and could perhaps be included there.*

<Author's response>
I have deleted the appendix. The following corresponding changes were also made.

<Author's changes in manuscript>
- The original Appendix A was deleted.
- l.100:
- Fig. 3 caption: "Celerity and low-pass filtered (subtidal)  background VM1 current speed , (b) …"

*Minor comments:*

*<Referee comments>*
*3. I think there is a sign error in the exponent of eq. (19a).*

<Author's response>
I have double-checked all the equations, and I think the sign is correct. Please note that tidal

phases are defined as phase lags (positive clockwise on the *x-y* plane, see Fig. 1) to be consistent with the traditional harmonic analysis throughout the manuscript. The following is intermediate steps in the calculation.

$$E(X_j + iY_j) = \int_0^\infty a f_A(a) da \times \frac{1}{\sqrt{2\pi}\sigma_j} \int_{-\infty}^\infty e^{-i\theta} \exp\left(-\frac{(\theta - \varphi_j)^2}{2\sigma_j^2}\right) d\theta$$

$$= \bar{a}_j \frac{e^{-i\varphi_j}}{\sqrt{2\pi}\sigma_j} \int_{-\infty}^\infty e^{-i(\theta - \varphi_j)} \exp\left(-\frac{(\theta - \varphi_j)^2}{2\sigma_j^2}\right) d\theta = \bar{a}_j e^{-\sigma_j^2/2} e^{-i\varphi_j}$$

\<Author's changes in manuscript\>
No change was made to the manuscript corresponding to this comment.

*\<Referee comments\>*
*4. First sentence below eq (20): State explicitly which of the components are deterministic and random.*

\<Author's response\>
Thank you for your comment. I have revised the sentence as follows.

\<Author's changes in manuscript\>
l. 203: "As seen in these relationships, and as shown before by Colosi and Munk (2006), the phase spread $\sigma_j$ provides a convenient way to separate the variance of each sinusoid $(\overline{a_j^2})$ into the deterministic (mean) component $(r_j^2)$ and the random (deviation) component $(\bar{a}_j^2 \varsigma_j^2)$."

*\<Referee comments\>*
*5. Second paragraph of section 3.3: I was not familiar with the word 'celerity'. As far as I have been able to find out, it is the same as phase velocity, but in the first sentence below eq (A3) it appears to be something else. Either replace 'celerity' by 'phase velocity', or explain the difference.*

\<Author's response\>
Thank you for your comment. I forgot defining celerity in Part I. There are different ways to define vertical modes, and some researchers include Coriolis effects in the calculation, which makes the eigenvalues the phase speeds of long linear internal waves. However, the definition

used in this study yields the phase speeds of non-rotating waves (i.e., $c_n$), which are not the phase speeds of internal tides. Since $c_n$ appears many times, I introduced a separate term to differentiate it from the phase speed of internal tides.

I have added the following sentence in the revised manuscript.

<Author's changes in manuscript>
l.273: "(In this study, the term "celerity" is deliberately used for the propagation speed of non-rotating, long, linear gravity waves with one of the vertical-mode structures, which differs from the phase speed of internal tides.)"

*<Referee comments>*
*6. Fig. 4: Stress that b is a result of the calculations, in contrast to σ_j and ϕ_j , which are inputs.*

<Author's response>
Thank you for your comment. I have revised the caption as follows.

<Author's changes in manuscript>
Figure 4 caption: "The $b$ parameter of the generalized Rayleigh distribution,  calculated from Eq. (6) and (9d), is shown in each panel."

*<Referee comments>*
*7. Section 4.2, second paragraph on page 21: I would expect the width of the spectral peaks in Fig. 5 to be related to the nonharmonic fraction. From the values for VM1 in Table 2, I would therefore expect the diurnal peak to be much narrower than the quarterdiurnal peak, but that is not the case. Please comment.*

<Author's response>
I have considered what determines the width of the "cusps" in the spectra, but I do not understand the causes. My expectation is that it depends on the time scale of the processes that modulate wave phases, because the width is the apparent change of wave frequency caused by phase modulation. So, I speculate that the width is independent of the height of the peak.

<Author's changes in manuscript>
No change was made to the manuscript based on this comment.

*<Referee comments>*

*8. Section 4.2, 4th sentence in the second paragraph on page 21: Is the first 'nonharmonic' a typo? I would think that the nonlinear interaction of harmonic semidiurnal tides can generate nonharmonic quarterdiurnal internal tides, which could explain why the latter has a much larger nonharmonic fraction.*

<Author's response>

Thank you for your comment. Because nonlinear interaction terms have the form (harmonic + nonharmonic) × (harmonic + nonharmonic), I would expect that the nonlinear interaction of harmonic–harmonic semidiurnal internal tides excites harmonic quarterdiurnal internal tides ($M_2+M_2=M_4$, $M_2+S_2=MS_4$, etc.) before modulation, and that the interaction of harmonic–nonharmonic or nonharmonic–nonharmonic semidiurnal internal tides excites nonharmonic quarterdiurnal internal tides ($M_2$+modulated $M_2$=modulated $M_4$, etc.) without modulation. This would make nonharmonic quarterdiurnal internal tides larger than the harmonic counterpart when the nonharmonic variance fraction of semidiurnal internal tides is ~0.5.

I have revised the following sentence.

<Author's changes in manuscript>

l.454: "This is partly expected because the nonlinear interaction of  harmonic–nonharmonic or nonharmonic–nonharmonic semidiurnal internal tides can generate nonharmonic quarterdiurnal internal tide without the modulation processes."

-------------------- Author's reply to RC2 --------------------
---------- General Comments ----------

*<Referee comments>*

*Internal tide observations consist of a harmonic part, well-modelled by a sum of sinusoids at tidal frequencies, and a nonharmonic part. The author models internal tide observations at a given frequency by the sum of many waves with random, time-dependent amplitudes and phases. The harmonic amplitudes and phases are given by the expectation value of the total complex amplitude of the wave, equivalent to a least-squares fit which is standard in literature. Subtracting this least-squares-fitted harmonic part from the total internal tide signal leaves the nonharmonic remainder.*

*By considering the internal tides at a single frequency to be the sum of many waves, the author allows for generation of tides from multiple sources, a novel contribution of this study. The waves propagate through the ocean, a random media, which justifies the random amplitudes and phases of the signal.*

*The author derives the total probability density functions (PDFs) of nonharmonic amplitude and phase from the individual amplitude and phase distributions. Statistical assumptions on the form of individual wave amplitude and phase PDFs allow for relatively straightforward computation of the total PDFs. These total PDFs compare favourably to observation (figure 6), seemingly justifying the author's multi-source, statistical approach.*

*The work is well-written and provides an interesting contribution to the field of nonharmonic-tide modelling. I am happy to recommend publication after the author has addressed the points below.*

<Author's response>
Thank you for your time and effort in reviewing this manuscript, and for your comments and suggestions. I have revised the manuscript following most of your suggestions. My point-by-point replies are listed below.

<Author's changes in manuscript>
Changes to the manuscripts are listed below my replies to your individual comments.

---------- Specific Comments ----------

*<Referee comments>*

*I am in support of the suggested alternative derivation by Jonas Nycander for section 2.1., providing the author agrees that the two are equivalent. (Please ignore points 2--5 below if the alternative derivation is implemented.)*

<Author's response>

I modified Section 2.1 following the suggestion by Dr. Jonas Nycander. The alternative derivation is not equivalent when the number of wave components (or sources) is small or moderate, which is required to show the convergence of the PDFs to the limiting distributions (Section 4.1). So, I moved the original equations to Appendix A.

<Author's changes in manuscript>
- l.136: New Section 2.1. "Probability distribution functions in many source limit" was added.
- l.596: Most of the original Sections 2.1 and 3.1 were moved to new Appendix A.

*<Referee comments>*

*Lines 147--8: For those of us who aren't statisticians, please can you state what the Fourier transform convention in statistics is?*

<Author's response>

Thank you for your comment. I studied statistics because it was necessary, but I am not a statistician either. In response to comments from another anonymous referee, I added a new appendix that summarizes the Fourier and Hankel transform pairs (Appendix B), and moved the sentence there. I modified the sentence as follows.

<Author's changes in manuscript>

l.713: "The signs of the exponents follow the statistical convention, which are different from the common definition (e.g., in physics and engineering)."

*<Referee comments>*

*Abramowitz and Stegun (cited line 158) has a digital successor, the Digital Library of Mathematical Functions (DLMF). This is a citable resource and much easier for the online reader to use. Please consider changing your reference to the relevant DLMF equations.*

<Author's response>

Thank you for your suggestion. I do use DLMF, and added reference to the DLMF equations.

(However, because I refer to papers that are more than 50 years old, and I appreciate their availability today, I kept reference to Abramowitz and Stegun because it is more persistent than online resources.)

<Author's changes in manuscript>
- l.178: "which comes from the so-called Jabobi-Anger expansion (Abramowitz and Stegun 1972, Eq. 9.6.34 or DLMF, Eq. 10.35.2)."
- l.643: "The second expression is obtained using the azimuthal Fourier series Eq. (B4a) (with $\phi^{(k)}$ being the Fourier coefficients) and the properties of the Bessel function (Abramowitz and Stegun 1972, Eqs. 9.1.5, 21, and 35 or DLMF, Eqs. 10.4.1, 10.9.2, and 10.11.1)."
- l.662: "the amplitudes $\alpha_{k,l}$ are obtained using the orthogonality of the Bessel function over a fixed interval (Abramowitz and Stegun, 1972, Eq.11.4.5 or DLMF, Eq.10.22.37):"
- l.728: "Using these azimuthal Fourier series and the so-called Jacobi-Anger expansion (Abramowitz and Stegun 1972, Eqs. 9.1.44 and 45 or DLMF, Eqs. 10.12.2 and 3) …"
References: DLMF was added.

*<Referee comments>*
*Eqn 9 can be obtained using just DLMF 10.9.2 and 10.4.1 rather than Eqs. 9.1.5, 35, 44, and 45 of Abramowitz and Stegun (line 158).*

<Author's response>
Thank you for your suggestion. I think the formula for negative argument (DLMF, Eq. 10.11.1) is still required. I added reference to DLMF Eqs. 10.4.1, 10.9.2 and 10.11.1, and revised the equation number for Abramowitz and Stegun. I have modified the sentence as follows.

<Author's changes in manuscript>
l.643: "The second expression is obtained using the azimuthal Fourier series Eq. (B4a) (with $\phi^{(k)}$ being the Fourier coefficients) and the properties of the Bessel function (Abramowitz and Stegun 1972, Eqs. 9.1.5, 21, and 35 or DLMF, Eqs. 10.4.1, 10.9.2, and 10.11.1)."

*<Referee comments>*
*Eqn 12: a reference to the Fourier-Bessel series used would be useful for the reader.*

<Author's response>
Thank you for your comment. Instead of adding a reference, I added more explanation of the Fourier-Bessel solution in response to comments from another anonymous referee, including

the assumed form of Fourier-Bessel series and the orthogonality required to calculate the amplitudes. I hope this makes the Fourier-Bessel solution more understandable.

<Author's changes in manuscript>
- l.653-671: New Eqs.(A7)-(A11) and related texts were added to explain the derivation of the Fourier-Bessel solution.
- l.705: New Appendix B was added to explain detailed points in the derivation of the Fourier-Bessel solution.

*<Referee comments>*
*Please define "celerity" or change to "phase speed" if the two are equivalent.*

<Author's response>
Thank you for your comment. I forgot defining celerity in Part I. There are different ways to define vertical modes, and some researchers include Coriolis effects in the calculation, which makes the eigenvalues the phase speeds of long linear internal waves. However, the definition used in this study yields the phase speeds of non-rotating waves (i.e., $c_n$), which are not the phase speeds of internal tides. Since $c_n$ appears many times, I introduced a separate term to differentiate it from the phase speed of internal tides. I added the following sentence in the revised manuscript.

<Author's changes in manuscript>
l.273: "(In this study, the term "celerity" is deliberately used for the propagation speed of non-rotating, long, linear gravity waves with one of the vertical-mode structures, which differs from the phase speed of internal tides.)"

*<Referee comments>*
*As I understand it, the author's vertical mode formulation cited as Shimizu (2017, 2019) on lines 282--3 and Shimizu (2011) on line 306 is somewhat equivalent to that of Kelly et al. (2012) and subsequent works. If this is the case, the author should cite the work of Kelly and co-authors as well.*

<Author's response>
I do not think I need to cite studies by Kelly and co-authors, because I do not use their formulation in my studies. The reasons why the formulations are somewhat equivalent (the use of horizontally variable vertical modes in the presence of large topography) are (i) because they

are based on previous studies for small topography (e.g., Llewellyn-Smith and Young 2002, Mauge and Gerkema 2008), and (ii) because Kelly and co-authors later adopted the formulation in Shimizu (2011). The formulation in Shimizu (2017, 2019) is based on the generalized isopycnal coordinate that allows the extension to full nonlinearity and nonhydrostaticity, unlike the height coordinate formulation in Kelly et al. (2016).

Followings are more details. Around 2010, Kelly et al. (2012) and I developed somewhat similar vertical mode formulation independently. Later, I noticed that similar formulation was already developed by Griffith and Grimshaw (2007), which I cited in Shimizu (2011). The record is clear that the evolutionary equations of modal amplitudes in the subsequent works by Kelly and co-authors (e.g., Coupled Shallow Water model, Kelly et al. 2016) are based on Shimizu (2011), because earlier works by Griffith and Grimshaw (2007) and Kelly and co-authors (Kelly et al. 2012, 2013, 2015) did not use the evolutionary equations, and because Kelly and co-authors were aware of Shimizu (2011) (cited in Kelly et al. (2012)). However, in the subsequent works, Kelly and co-authors removed reference to Shimizu (2011) and cited only Griffith and Grimshaw (2007). I hope you would tell Kelly and co-authors to acknowledge the originality of my work and to cite my study appropriately in the future.

<Author's changes in manuscript>
No change was made to the manuscript based on this comment.

<Referee comments>
*The author should mention the SWOT mission in his introduction (e.g. Morrow et al. 2019), given it is a key reason for current interest in the nonharmonic tide.*

<Author's response>
Thank you for your suggestion. I added the following sentence in the revised manuscript.

<Author's changes in manuscript>
- l.540: "These parameters provide important bases for distinguishing quasi-geostrophic (or "balanced") currents and internal tides in wide-swath altimeter data, such as those from Surface Water and Ocean Topography (SWOT) mission (Morrow et al. 2019)."
- References: Morrow et al. (2019) was added.

<Referee comments>

*Kachelein et al. (2024) consider amplitude modulation for nonharmonic tides in the California Current, where tides propagate from many generation sites. Perhaps this relevant work should be cited?*

<Author's response>

Thank you for the suggestion. Kachelein et al. (2024) is relevant to this study. I added the following sentence in the revised manuscript.

<Author's changes in manuscript>

- l.67: "[After preparing the original manuscript of this paper and presenting the selected results at Ocean Sciences Meeting 2024 (Shimizu, K., Developing a statistical model of incoherent internal tides, 19--23 February 2024), I became aware of Kachelein et al. (2024), who showed the PDF of non-phase-locked internal tides from high-frequency radar observations.]"
- References: Kachelein et al. (2024) was added.

---------- Technical Comments ----------

*<Referee comments>*
*- Typo line 35 immediately before barotropic. Inverted comma (') should be quotation marks (").*
*- Typo line 91. "mooing site" should be "mooring site".*
*- Line 215: missing "the" after "We also need".*
*- Grammar lines 375--6. Could be resolved with the addition of "for" i.e. "searched for numerically."*
*- Clunky phrasing lines 480--1: "However, this difference does not matter to see that the phase is roughly uniformly distributed". Something like "However, the phase is roughly uniformly distributed despite this difference" would work better.*
*- Change "would be" on lines 521 and 522 to "is".*

<Author's response>

Thanks you for finding errors and your suggestions. Sentences were corrected or modified as you suggested.

<Author's changes in manuscript>

- l.35: 'barotropic' -> "barotropic"
- l.94: "mooing site" -> "mooring site"
- l.211 "the" was added before "amplitude distribution f_A"

- l.369: "for" was added before "numerically"

- l.474: The sentence was revised as "However, this difference does not matter to see that the phase is roughly uniformly distributed (small b parameters) despite this difference."

- l.515-516: Two instances of "would be" were changed to "is"

-------------------- Author's reply to RC3 --------------------
---------- General Comments ----------

*<Referee comments>*

*This paper derives the probability distribution of a sum of waves, where the amplitudes and phases of the component waves in the sum are assumed to be random variables. The development sets down a clear derivation in one place. It also does a nice job of contextualizing and motivating the derivation as a continuation of previous work. The author provides a clear justification for the term "nonharmonic", in contrast to other terms, "non-phase-locked", "incoherent", and "non-stationary", to describe the type of tidal signals which should be described by the proposed probability distribution. Finally, the author provides a good qualitative discussion of the distribution in the limit of multiple wave sources with different properties, which emphasizes the rate at which the asymptotic distribution is approached, and provides insight into how this tends to obscure information about the underlying properties of the component waves. For all these reasons, I think the paper makes a nice contribution to the field, and it ought to be published.*

*Although I am familiar with the basic methods employed in the article, it would take me considerable time to verify all the steps of the derivation in detail, and I have not attempted to do so. In order to increase the pedagogical value of the paper, and also to instill more confidence in the results, I would encourage the author to include more detail, possibly in the form of additional short appendices.*

    <Author's response>
    Thank you for your time and effort in reviewing this manuscript, and for your comments and suggestions. I have revised the manuscript following your suggestions. Based on the comments from the other referees, I have moved most of Sections 2.1 and 3.1 to new Appendix A, and added further details in Appendix A and new Appendix B based on your comments. My point-by-point replies are listed below.

    <Author's changes in manuscript>
    Changes to the manuscripts are listed below my replies to your individual comments.

---------- Minor Comments ----------
*<Referee comments>*
*l113: It is not clear to me what the expected value is averaging over. Is it assumed that the random realizations represent different time series? Or will this be used with segments of time series of*

*short duration, or what, exactly? (Oh, I see later --- the segments of the time series are treated as independent realizations.)*

<Author's response>

Statistical theory differentiates the (true) expectation value from sample mean, and the expectation value in l.113 is the theoretical true value. In the analysis of observational data, I assumed that the segments separated longer then the decorrelation time (Table 1) are statistically independent samples.

Judging from the comments from the other reviewers, I am afraid this is a detail that many oceanographers find distracting. So, I did not modify the manuscript based on this comment.

<Author's changes in manuscript>

No change was made to the manuscript based on this comment.

*<Referee comments>*

*l142-143: Can he put these steps into an appendix?*

<Author's response>

Based on comments from the other referees, I have moved Section 2.1 to new Appendix A, and added the intermediate steps there. Please see new Eq. (A1) and texts around it.

<Author's changes in manuscript>

- l.618-625: The steps are explained in these new sentences and new Eq. (A1).

- l.705: New Appendix B was added to show Fourier and Hankel transform pairs in Cartesian and polar coordinates.

*<Referee comments>*

*l149: ¥phi is the characteristic function of A',¥Theta' ?*

<Author's response>

Thank you for your comment, and I am sorry that this was not clear. It is the characteristic function of either $f_{A'\Theta'}$ or $f_{X'Y'}$, because I assumed that the "wavenumber" vector used in Fourier transform follows the standard rule of coordinate transformation. I added Appendix B based on this and your other comments, and added the following sentence.

<Author's changes in manuscript>
l.722: "Note also that, unlike the PDF, we do not distinguish $\phi$ in Cartesian and polar coordinates, and the "wavenumber" vector follows the standard rule of coordinate transformation."

*<Referee comments>*
*l166-l171: Sorry, I'm not sure how to check these.*

<Author's response>
Thank you for your comment, and I am sorry that I omitted the details. It involves the comparison of Fourier-Bessel series and Hankel transform, which is a technique used by e.g., Bennett (1948) and Barakat (1974, 1988). I added the derivation of the Fourier-Bessel solution in Appendix A, Eq.(A7)-(A11). This required defining Hankel transform pair, which is described in new Appendix B.

In order to make the derivation of the Fourier-Bessel solution cleaner, I have changed the sign of azimuthal Fourier series (now Eq. (B4a)), which changed the sign in front of the imaginary units in Eqs. (A4) and (A7). They do not change the results of this study (which is real valued).

<Author's changes in manuscript>
- l.653-671: New Eqs.(A7)-(A11) and related texts were added to explain the derivation of the Fourier-Bessel solution.
- l.705: New Appendx B was added to explain detailed points in the derivation of the Fourier-Bessel solution.
- Eqs. (A4), (A7), and (B4a): the signs in front of the imaginary units were changed.

*<Referee comments>*
*l181-l182: So are these the transformation of the joint Gaussian in Cartesian coordinates to the polar coordinates?*

<Author's response>
Yes. I have simplified the derivation in Section 2.1 following suggestions by the other referees. I hope this is clear now.

<Author's changes in manuscript>
l.164-170: These new sentences explain the transformation.

*<Referee comments>*
*l190: Why is the phase distribution bimodal?*

<Author's response>

Because the contour of the bivariate joint Gaussian distribution has the shape of an ellipse, whose major axis lies in the direction of, say, ¥theta and ¥theta+¥pi. I have simplified the derivation in Section 2.1 following suggestions by the other referees, and I hope this point is clear now.

<Author's changes in manuscript>

l.164-170: These new sentences explain the transformation from the joint Gaussian to bimodal phase distribution.

*<Referee comments>*
*l276: "tidal" --> "non-tidal"?*

<Author's response>

No, tidal variability is removed to calculate background elevation.

<Author's changes in manuscript>

No change was made to the manuscript based on this comment.

*<Referee comments>*
*l306: I'm not sure about the units here, given the prior normalizations.*

<Author's response>

I am sorry that the units were unclear. I have added / revised the following sentences listed below. Then, the units of $P\_n$ and $K\_n$ are J m^{-2}, and that of $J\_n$ is W m^{-1}, as in Table 2. (The scaling Eq. (17) does not change the unit.)

<Author's changes in manuscript>

- l.277: "In this paper, the vertical modes were normalized by setting the arbitrary norm for the barotropic mode (h_0) to the water depth (201 m), and the norms for VM1 (h_1), VM2 (h_2), VM3 (h_3), and VM4 (h_4) to 1/5, 1/17, 1/38, and 1/63 of the water depth, respectively."
- l.284: "(The units of ¥eta_n and v_n are m and m s^{-1}, respectively.)"

*<Referee comments>*
*l319: "displaement"*

    <Author's response>
    Thank you. Corrected.

    <Author's changes in manuscript>
    l.312: "displaement" -> "displacement"

*<Referee comments>*
*l385: Is this the same as Colosi and Munk?*

    <Author's response>
    No. Since Colosi and Munk did not derive the limiting PDFs, they did not do this calculation.

    <Author's changes in manuscript>
    No change was made to the manuscript based on this comment.

*<Referee comments>*
*l390: Not sure I understand. Couldn't it be estimated using the same approach as for fitting the Lorentzians?*

    <Author's response>
    Thank you for your comment. Yes, it could be estimated as you suggested, but I did not do so because the contribution of the background level was relatively small (compare of background level in Table 1 and potential energy in Table 2). I have revised the sentence as follows.

    <Author's changes in manuscript>
    l.382: "Note also that the background level in the PSD is included in $\sigma_{A'}^2$ estimated in this way, but excluded in $\sigma_{A'}^2$ estimated by fitting the double Lorentzian model."

*<Referee comments>*
*l70: these are all good justifications*

*l83: Good---it looks like he has familiarized himself with the literature of other fields.*

*l224-232: good qualitative discussion*

*l415-l423: good qualitative discussion*

<Author's response>
Thank you.

<Author's changes in manuscript>
No changes were made to the manuscript based on these comments.

-------------------- References --------------------

Griffiths, S. D., and R. H. J. Grimshaw, 2007: Internal tide generation at the continental shelf modeled using a modal decomposition: Two-dimensional results. J. Phys. Oceanogr., 37, 428–451, doi:10.1175/JPO3068.1.

Kelly, S. M., P. F. J. Lermusiaux, T. F. Duda, and P. J. Haley, 2016: A Coupled-Mode Shallow-Water Model for Tidal Analysis: Internal Tide Reflection and Refraction by the Gulf Stream. J. Phys. Oceanogr., 46, 3661–3679, doi:10.1175/JPO-D-16-0018.1.

Kelly, S. M., J. D. Nash, K. I. Martini, M. H. Alford, and E. Kunze, 2012: The cascade of tidal energy from low to high modes on a continental slope. J. Phys. Oceanogr., 42, 1217–1232, doi:10.1175/ JPO-D-11-0231.1.

Kelly, S. M., N. L. Jones, and J. D. Nash, 2013: A coupled model for Laplace's tidal equations in a fluid with one horizontal dimension and variable depth. J. Phys. Oceanogr., 43, 1780–1797, doi:10.1175/JPO-D-12-0147.1.

Kelly, S. M., N. L. Jones, G. I. Ivey, and R. N. Lowe, 2015: Internal tide spectroscopy and prediction in the Timor Sea. J. Phys. Oceanogr., 45, 64–83, doi:10.1175/JPO-D-14-0007.1.

Llewellyn-Smith, S. G., and W. R. Young, 2002: Conversion of the barotropic tide. J. Phys. Oceanogr., 32, 1554–1566.

Mauge, R., and T. Gerkema, 2008: Generation of weakly nonlinear nonhydrostatic internal tides over large topography: A multi-modal approach. Nonlinear Processes Geophys., 15, 233–244.

---

## Author Response (AR2)

Dear Dr. Julian Mak,

Thank you for reading and handling the manuscript. I have revised the manuscript following the referee's and your comments. The details are listed below.

I am looking forward to hearing the outcome of the manuscript.

Kind regards,
Kenji Shimizu

*<Editor comments>*
*As discussed, would suggest the competing statement be shortened and rephrased accordingly (not that bothered about the change of affiliation). Would suggest for it simply being "KS is employed as a consultant in Australia, and involved in commercial projects related to the topic of this paper."*

<Author's response>
I have revised the competing interests statement as you suggested.

<Author's changes in manuscript>
Competing Interests section: "KS is employed as a consultant in Australia, and involved in commercial projects related to the topic of this paper. ~~KS has a competitive relationship with some physical oceanographers at University of Western Australia for industry-funded projects on topics related to this paper. During the review stage, University of Western Australia requested through his employer that KS removes the statement regarding the competing relationship, and names a collaborative project between University of Western Australia and his employer, even though this study received no funding or indirect support from these parties or the project. KS declined the requests, but it required the removal of the name of his employer from this paper.~~"

*<Referee comments>*
*On line 228 it is written: 'the e-folding standard deviation (where the dashed line reaches 1 in Fig. 2d) is 16% of the full phase $2\pi$'. Please explain. What does the sloping dashed line represent, and how does the e-folding standard deviation relate to this plot? Please also improve the description of Fig. 2b in the figure caption. Perhaps it would be more logical to*

*replace the horizontal dashed line by a solid line, which would also make the sentence cited above less ambiguous?*

<Author's response>
I thank the referee's comments. The following changes were made to the revised manuscript.

<Author's changes in manuscript>
l.219: "For example, the e-folding  variance (where the dashed line reaches 1 in Fig. 2d) is about 1, or 16% of the full phase $2\pi$ in terms of standard deviation."
Fig. 2: The horizontal line in panel (d) was changed from a dashed line to a dashed double-dotted line.
Caption to Fig. 2: "(d) normalized contributions to E(A'^2) (first term (dashed double-dotted line) and second terms (solid line) in Eq. (14b)."